# Temporal rainfall disaggregation using a micro-canonical cascade model: Possibilities to improve the autocorrelation

Hannes Müller-Thomy[1,*]

[1]Institute of Hydraulic Engineering and Water Resources Management, Vienna University of Technology, Karlsplatz 13/222, 1040 Vienna, Austria

[*]previously published under the name Hannes Müller

*Correspondence to*: Hannes Müller-Thomy (mueller-thomy@hydro.tuwien.ac.at)

**Abstract.** In urban hydrology rainfall time series of high resolution in time are crucial. Such time series with sufficient length can be generated through the disaggregation of daily data with a micro-canonical cascade model. A well-known problem of time series generated so is the inadequate representation of the autocorrelation. In this paper two cascade model modifications are analysed regarding their ability to improve the autocorrelation in disaggregated time series with 5 minute resolution. Both modifications are based on a state-of-the-art reference cascade model (method A). In the first modification, a position-dependency is introduced in the first disaggregation step (method B). In the second modification the position of a wet time step is redefined in addition (method C). Both modifications led to an improvement of the autocorrelation, especially the position redefinition. To ensure the conservation of a minimum rainfall amount in the wet time steps, the mimicry of a measurement device is simulated after the disaggregation process. Simulated Annealing as a post-processing strategy was tested as both, as an alternative as well as an addition to the modifications in method B and C to improve the autocorrelation. For the resampling, a special focus was given to the conservation of the extreme rainfall values. Therefore, a universal extreme event definition was introduced to define extreme events a priori without knowing their occurrence in time or magnitude. The resampling algorithm is capable of improving the autocorrelation, independent of the previously applied cascade model variant. Also, the improvement of the autocorrelation by the resampling was higher than by the choice of the cascade model modification. The best overall representation of the autocorrelation was achieved by method C in combination with the resampling algorithm. The study was carried out for 24 rain gauges in Lower Saxony, Germany.

## 1. Introduction

For many applications in hydrology high resolution rainfall time series are crucial (see the review of Cristiano et al. (2017)) to match the scale of the underlying processes (Blöschl and Sivapalan, 1995). Schilling (1991) concludes that for urban hydrology, in particular for overland flow, a temporal resolution of 5 minutes is acceptable. Berne et al. (2004) points out

that the required temporal resolution depends on the catchment size, and recommend for urban catchments with area sizes of about 1000 ha a temporal resolution of 6 minutes and for 10 ha or smaller a temporal resolution of 1 min. Unfortunately, lengths of time series with such a high temporal resolution are insufficient for most applications. However, for the non-recording stations (registration of daily values) the time series lengths are usually sufficient, but the temporal resolution is not fine enough to cope with the dynamics in urban hydrology (Ochoa-Rodriguez et al., 2015) or for erosion processes (Jebari et al, 2012).

A possible solution for this data scarcity is rainfall disaggregation. Information of short, high-resolution time series are used to disaggregate coarser time series. The disaggregation results in high-resolution time series with sufficient lengths as well as a higher network density in most cases. Several methods exist for the temporal disaggregation, e.g. method of fragments (Wójcik and Buishand, 2003, Westra et al., 2012, Breinl et al., 2015, Breinl and Di Baldassarre, 2019), rectangular pulse models (Koutsoyiannis and Onof, 2001) and cascade models. Cascade models are well-known disaggregation models for the generation of high-resolution rainfall time series and were developed originally in the field of turbulence theory (Mandelbrot, 1974). Based on Koutsoyiannis and Langousis (2011), the canonical version of the cascade model (conservation of rainfall amount on average during the disaggregation, e.g. Molnar and Burlando, 2005, Paschalis et al., 2012) represents a downscaling technique, while the micro-canonical version (exact rainfall amount conservation for each time step, e.g. Olsson, 1998, Güntner et al., 2001, Licznar et al., 2011, 2015, Müller-Thomy et al., 2018, Müller-Thomy-Sikorska-Senoner, 2019) represents a disaggregation technique. However, for urban hydrology the majority of investigations with cascade models focus on the disaggregation of quasi-daily time series (with time step durations of 1280 minutes instead of 1440 minutes, e.g. Licznar et al., 2011, 2015, Molnar and Burlando, 2005, Paschalis et al., 2014) to achieve a final temporal resolution of 10 minutes or 5 minutes. This enables using the same branching number (that determines the number of finer time steps with equal duration resulting from one coarser time step) of $b=2$ throughout the disaggregation process with intermediate resolutions of {640, 320, 160, 80, 40, 20, 10, 5 min}. Since time series with 1280 minutes do not exist as observations, these studies are rather theoretical than practical from an engineering point of view. Of course, by the application of an additional transformation process a desired temporal resolution can be achieved, whereby the transformation process affects the characteristics of the disaggregated time series. To overcome this issue, Müller and Haberlandt (2018) developed a micro-canonical cascade model, which enables the rainfall disaggregation from daily values to 5 minutes. Müller and Haberlandt evaluated the disaggregated rainfall time series in terms of rainfall characteristics and showed good performances regarding continuous (average intensity, fraction of dry intervals) and event-based rainfall characteristics (wet and dry spell duration, wet spell amount) as well as extreme values. An additional validation with an urban hydrological model led to comparable results for event-based combined sewer overflow volume as well as manhole flooding volume when forced with observed and disaggregated rainfall time series, respectively.

However, Müller and Haberlandt (2018) also show that the autocorrelation of the disaggregated time series is underestimated. This is critical, because the autocorrelation describes the memory of a process. So for continuous

applications especially deviations can be expected whether e.g. an urban hydrological model is forced with observed or disaggregated rainfall time series. The underestimation of the autocorrelation in the generated time series has been identified before when micro-canonical cascade models were used for the disaggregation by e.g. Olsson (1998), Güntner et al. (2001) and Paschalis et al. (2012, 2014). Lisniak et al. (2013) divided the study period into a calibration and validation period.

While for the calibration period the autocorrelation was underestimated, a good representation was achieved for the validation period. Rupp et al. (2009) and Pohle et al. (2018) analysed four and three different kinds of cascade models, respectively. Depending on the choice of the model, under- and overestimations of the autocorrelation function were identified. A good representation of the lag-1 autocorrelation was achieved by Hingray and Ben Haha (2005) with two micro-canonical cascade models. However, since only four disaggregation steps were applied (from hourly to 7.5 min time

steps) it remains unclear, if the good representation would have been achieved for more disaggregation steps.

Summarizing the previous findings, an adequate reproduction of the autocorrelation function by the multiplicative micro-canonical cascade model can be difficult. The reasons for over- and underestimation differ depending on the choice of the cascade model. For example, in Olsson (1998) and Müller and Haberlandt (2018) the underestimation is caused by the generation of dry time steps inside rainfall events, causing shorter wet spells in the disaggregated time series in comparison

with the observed time series. In Pohle et al. (2018) an overestimation of the autocorrelation is identified for a cascade based on Menabde and Sivaplan (2000), which disables the generation of dry finer time steps from one wet coarser time step.

In this study, I investigate modifications of the cascade model itself, but also a post-processing strategy after the disaggregation procedure to improve the representation of the autocorrelation in the disaggregated time series. The basis for all investigations in this study is the multiplicative random cascade model as proposed by Müller and Haberlandt (2018).

According to Marshak et al. (1994) it is a bounded cascade model with a single parameter set for each disaggregation level. The parameters are estimated by the aggregation of observed, high-resolution time series (Carsteanu and Foufoula-Georgiou, 1996). The modifications are based on the introduction of position-dependencies with two different degrees of complexity. The first, less complex modification includes taking into account the position of the wet day in the time series. The second, more complex modification follows an idea of Lombardo et al. (2012, 2017). Lombardo et al. analysed which time steps are

most worth to consider to generate highly correlated time series under the burden of computational efforts. Their conclusion is adapted in this investigation. Both modifications are expected to improve the autocorrelation function and lead with the basis model to three different cascade model variants in this study. It should be pointed out that Lombardo et al. (2012) showed that the disaggregation process of discrete multiplicative random cascade models is non-stationary, "because the autocorrelation structure depends on the position in time in an undesirable manner". Since the aim of this study is to improve

the overall representation of the autocorrelation function (as average over time), it is not analysed to which extent the non-stationarity issue is solved by the introduced methods.

Simultaneously, a second general issue of the cascade model is solved: the generation of time steps with very small rainfall intensities. Molnar and Burlando (2005) identified a fraction of rainfall intensities lower than the measurement resolution of the investigated time series of 48 % for 10 min time series, starting with the disaggregation from quasi-daily values. Müller and Haberlandt (2015) found for a disaggregation from daily to hourly values a fraction of underestimated rainfall intensities of 35 %. Koutsoyiannis et al. (2003) argue that it is unclear, if the values generated by the cascade model are too small in comparison to the observed minimum intensities or if the resolution of the measurement device is not fine enough to observe the very small rainfall intensities generated by the cascade model. From a practical point of view, these low-intensity time steps are not important, but they have an impact on the autocorrelation function. To enable comparisons between the autocorrelation of observed and disaggregated rainfall time series a novel method is applied in this study to ensure a minimum rainfall intensity in the disaggregated time series.

In addition to the modifications of the cascade model itself, a resampling algorithm as post-processing strategy is analysed to improve the autocorrelation. A similar approach has been investigated by Bárdossy (1998), who used a simulated annealing algorithm to resample time series generated with a Markov chain Monte Carlo method. Bárdossy investigated temporal resolutions of 1 hour and 5 minutes, the autocorrelation function could be reproduced well for both. For this investigation, the proposed resampling algorithm of Müller and Haberlandt (2015) will be modified to include the autocorrelation function in the optimization process.

As a summary from the introduction, the main research question of this study is: How can the autocorrelation in the disaggregated time series be improved?

## 2. Rainfall data

For the investigation 24 stations in and around Lower Saxony, Northern Germany, are used (see Fig. 1). The same data set has been used before by e.g. Callau and Haberlandt (2017) for rainfall generation.

There are three dominating topographical regions with a coastal area around the North Sea, followed by the flatland around the Lüneburger Heide and the Harz middle mountains with altitudes up to 1141 m a.s.l. (from North to South).

Due to the climate classification after Köppen-Geiger (Peel et al., 2007) the study area can be divided into a temperate climate in the north and a cold climate in the mountainous region. Both climates exhibit no dry seasons, but hot summers. For the Harz mountains, average annual precipitation amounts greater than 1400 mm can be identified.

In Fig. 1, the 24 recording stations operated by the German Weather Service (DWD) with long term time series ranging from 9 to 20 years and a temporal resolution of 5 minutes are shown. The validation of the cascade model modifications is based on these 24 stations with a focus on the autocorrelation, but also on overall characteristics (average intensity, fraction of dry hours), event characteristics (dry spell duration, wet spell duration, wet spell amount) as well as extreme values. The definition for a single event is according to Dunkerley (2008); having a minimum of one dry time step before and after the

rainfall occurrence. For the definition of a dry time step the accuracy of the measuring instrument is not applied here, instead a threshold of 0 mm/5 min rainfall intensity is used. This enables comparisons between observed and disaggregated time series, which are not limited to the measuring accuracy and hence also includes smaller values (Molnar and Burlando, 2005). The rainfall time series characteristics along with further information of the rain gauges are provided in Table 1.

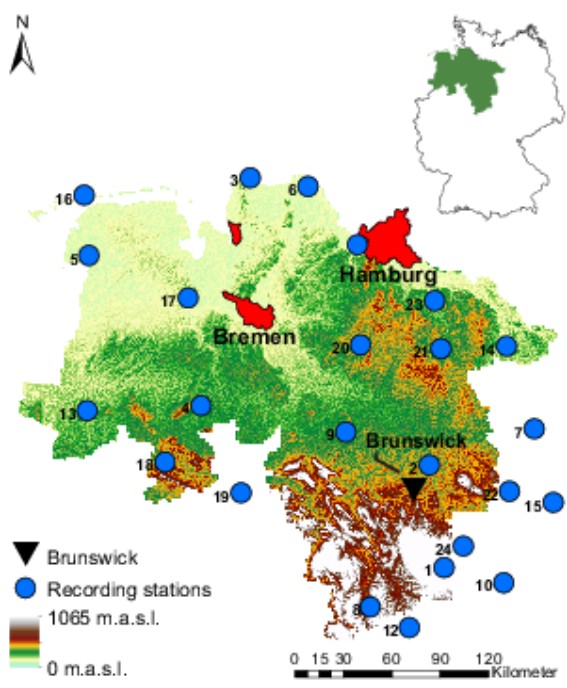

Fig. 1: Study area of Lower Saxony (and its location in Germany) with 24 recording stations.

## 3. Methods

The overall aim of this investigation is the improvement of the autocorrelation $r_{t1,t2}$ of the generated time series with a temporal resolution of 5 minutes. The autocorrelation function (Eq. 1) describes the memory of a process (here: rainfall) by the comparison of time series with itself in the future (shifted time series), whereby the future is represented by a certain number of $\lambda$ future time steps in the time series (lags). For rainfall time series Pearson's autocorrelation and Spearman's rank autocorrelation are applied in literature (e.g. Pohle et al., 2018). Actually, the Pearson's autocorrelation can only be applied if the data population is normally distributed, while Spearman's rank autocorrelation demands no assumption about the distribution of the data. However, using Pearson's autocorrelation has two advantages: i) it enables comparisons with results

from the literature, since it is applied more often than Spearman's rank autocorrelation, and ii) in terms of the later introduced resampling as optimization algorithm (see Sect. 3.2) for Pearson's autocorrelation no rank analysis of the whole time series has to be performed, since it can be calculated straight forwardly from the absolute values of the time series, which essentially fastens the optimization. Hence, the Pearson's autocorrelation is applied throughout this study.

5 The autocorrelation function is based on two elements: the covariance $s_{t1,t2}$ of the original and the shifted time series ($t_1$ and $t_2$, shifted by the lag $\lambda$ with $\lambda=t_2-t_1$), that describes the relation of both time series, and the standard deviations of both time series, $s_{t1}$ and $s_{t2}$, for the standardization of the covariance. While $t_1$ consists of $n$ time steps, the rainfall amount at a single time step $i$ is represented by $x$.

$$r_{t1,t2} = \frac{s_{t1,t2}}{s_{t1}s_{t2}} = \frac{\sum_{i=1}^{n-\lambda}(x_i-\overline{x})(x_{i+\lambda}-\overline{x})}{\sqrt{\sum_{i=1}^{n-\lambda}(x_i-\bar{x})^2 \sum_{i=1}^{n-\lambda}(x_{i+\lambda}-\bar{x})^2}} \tag{1}$$

10 To improve the autocorrelation of the disaggregated time series, several methods are introduced. A flowchart of the applied methods is presented in Fig. 2. The method chapter is divided into three subsections, which will be briefly described. Section 3.1 includes the model description of the cascade model and two modifications to improve the representation of the autocorrelation in the disaggregated time series. These three cascade model variants are compared at the end of Section 3.1. In Sect. 3.2 a resampling algorithm to increase autocorrelation as a post-processing strategy after the disaggregation process 15 is explained. The evaluation strategy for the disaggregated time series based on rainfall characteristics is explained in Sect. 3.3.

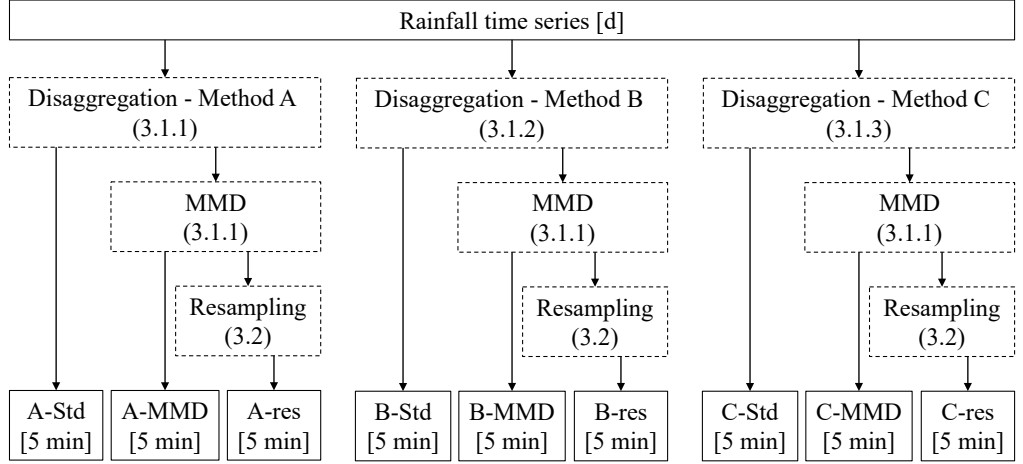

**Fig. 2: Overview of applied methods (dashed rectangles) and the resulting data sets (bottom line). In the brackets** 20 **behind the applied methods the subsection with the method description is referenced.**

### 3.1 Disaggregation model

### 3.1.1 General scheme (Method A)

The principle of the micro-canonical, bounded cascade model applied in this investigation is illustrated for the first two disaggregation steps in Fig. 3 (top) and was introduced by Müller and Haberlandt (2018). A coarse time step is disaggregated
into $b$ finer time steps, with $b$ named the branching number.

Starting with daily values, $b=3$ is applied and three time steps with 8 h duration are generated (similar to Lisniak et al., 2013). The choice of $b=3$ has no physical reason and is only applied to achieve a final temporal resolution of 5 min (see Section 1). The daily rainfall amount can occur in only one (0/0/1-splitting), in two ($0/\frac{1}{2}/\frac{1}{2}$) or in all three of the finer time steps, whereby the rainfall amount is distributed uniformly on the wet time steps (as it can be seen from the numbers in
brackets that identify the fractions of the daily rainfall amount). The required parameters for this splitting are the probabilities $P$ for one ($P(0/0/1)$), two ($P(0/\frac{1}{2}/\frac{1}{2})$) and three ($P(\frac{1}{3}/\frac{1}{3}/\frac{1}{3})$) wet 8 h-intervals in a day. The parameters $P(0/0/1)$ and $P(0/\frac{1}{2}/\frac{1}{2})$ have no influence on the position of the wet boxes, only on the number. The position is assigned randomly. The probability for $P(\frac{1}{3}/\frac{1}{3}/\frac{1}{3})$ can be determined by $P(\frac{1}{3}/\frac{1}{3}/\frac{1}{3}) = 1-P(0/0/1)-P(0/\frac{1}{2}/\frac{1}{2})$. The possible splittings and the distribution of the daily rainfall amount on the finer time steps are summarized in the cascade generator for $b=3$ (see Eq. 2). By multiplying the
rainfall volume $V$ of the coarser time step with the so-called multiplicative weights $W_1$, $W_2$ and $W_3$ the rainfall amounts of the finer time steps are derived. The sum of the weights is equal to 1 in each split, so the rainfall amount is conserved exactly throughout the disaggregation process.

$$W_1, W_2, W_3 = \begin{cases} \{1,0,0; \ 0,1,0 \ or \ 0,0,1\} & with \ P(0/0/1) \\ \{\frac{1}{2},\frac{1}{2},0; \ \frac{1}{2},0,\frac{1}{2} \ or \ 0,\frac{1}{2},\frac{1}{2}\} & with \ P(0/\frac{1}{2}/\frac{1}{2}) \\ \frac{1}{3},\frac{1}{3},\frac{1}{3} & with \ P(\frac{1}{3}/\frac{1}{3}/\frac{1}{3}) \end{cases} \tag{2}$$

Also, a volume-dependency of the parameter was identified for $b=3$. Müller and Haberlandt (2015) have shown that for days with high rainfall amounts (above a quantile $q_{0.998}$) the probability for two and especially three wet 8 h time steps is much higher than for lower daily rainfall amounts. Without a consideration of this volume-dependency of the parameters, the probability is too high that high daily rainfall amounts are put into one 8 h time step, which will lead to an overestimation of
extreme rainfall values. Hence, parameters are estimated for a lower and an upper volume class, with the quantile $q_{0.998}$ of all daily total rainfall amounts as threshold (see Müller and Haberlandt (2015) for more details).

After the first disaggregation step, only $b=2$ is applied. The generated intermediate time series have temporal resolutions of $\Delta t=\{4\ h,\ 2\ h,\ 1\ h,\ 30\ min,\ 15\ min,\ 7.5\ min\}$. The rainfall amount of the coarser time step can be assigned either to the first (1/0-splitting) or to the second (0/1) finer time step only or to both finer time steps (x/(1-x)). Again, all probabilities (P(0/1),

P(1/0) and P(x/(1-x))) sum up to 1. For the x/(1-x)-splitting, the relative fraction of the rainfall volume that is assigned to the first of the two finer time steps is considered as a random variable $x$ with $0 < x < 1$. An empirical distribution function is used to represent $f(x)$, with a maximum of 14 equidistant bins (based on the number of available splittings, see Storm (1988, p. 86)). The cascade generator for $b=2$ is given in Eq. 3:

$$W_1, W_2 = \begin{cases} 0 \text{ and } 1 & \text{with } P(0/1) \\ 1 \text{ and } 0 & \text{with } P(1/0) \\ x \text{ and } 1-x & \text{with } P(x/(1-x)); \ 0<x<1 \end{cases} \qquad (3)$$

The parameters for the splitting with $b=2$ depend on both, the position and the volume class of the current time step to disaggregate in the time series (see e.g. Olsson, 1998, Güntner et al., 2001). The position of a time step is defined by the wetness state of the surrounding time steps, so starting (time step before is dry, time step afterwards is wet, dry-wet-wet), enclosed, ending and isolated positions can be distinguished (see Fig. 4 for an illustration). For each position two volume

classes are defined, whereby the lower and upper volume class are separated by the mean rainfall volume of each position.

All parameters for $b=2$- and $b=3$-splittings can be estimated by aggregating observed time series with the same temporal resolution (Carsteanu and Foufoula-Georgiou, 1996). As mentioned before, a bounded cascade model is applied (Marshak et al., 1994). In bounded cascade models the weights $W$ depend on the temporal resolution to allow the disaggregation process to become smoother with finer resolutions (e.g. Menabde et al., 1997, Lombardo et al., 2012). The need for particular

parameter sets for each temporal resolution arises from the wide temporal range (from daily to 5 min time steps) and hence the underlying processes, which differ between the temporal scales.

To summarize the previous explanation regarding parameter estimation: for each temporal scale two fine time steps are aggregated (or three finer time steps for $b=3$, respectively) to one coarser time step, whereby the position and the volume class of the coarser time step determines to which position-volume class-combination the current splitting belongs. The

cascade model parameters are then estimated over all splittings of a position-volume class combination, so all parameters and distribution functions included in the disaggregation process are estimated empirically.

A final temporal resolution of 5 min is achieved via uniform transformation (Müller and Haberlandt, 2018). The rainfall amounts of time steps with $\Delta t=7.5$ min are distributed uniformly on 2.5 min time steps and afterwards aggregated non-overlapping to $\Delta t=5$ min.

As mentioned in Sect. 1, the cascade model tends to generate too many time steps with too low intensities. To overcome the issue, the behaviour of a measurement device is imitated after the disaggregation. Rainfall amounts smaller than the minimum resolution of the measurement device are summated in the chronological order of the time steps until the sum $S_{thr}$ exceeds this threshold. The former wet time steps with smaller intensities are set to 0 mm, while $S_{thr}$ is moved to the last time step of the summation. Afterwards, $S_{thr}$ is set back to 0 mm. This process is carried out over the whole disaggregated time

series and is referred to as 'mimicry of a measurement device' (MMD). If applied, $S_{thr}$ is set to 0.01 mm in this study, which is identical to the minimum resolution of the measuring devices of the observed time series.

### 3.1.2 Introduction of position-dependency in the uniform splitting (Method B)

For method B, only the first disaggregation step is (uniform splitting) modified from Method A in terms of the introduction of a position dependency. All further disaggregation steps remain identical to Method A and are not changed.

For the disaggregation of daily values into 8 hours the cascade model is applied with a branching number of *b=3*. Although the number of wet 8 hour-intervals depends on estimated probabilities, their position is chosen randomly in Method A. This is assumed to cause deviations of the autocorrelation.

Therefore, in addition to the volume classes, the position of the daily time step in the time series is also taken into account for the parameter estimation in Method B. The same positions are applied for the further disaggregation steps with *b=2* (starting, enclosed, ending and isolated position, see also Fig. 4). For each position, the probability of possible placements of wet and dry 8 hour-intervals is estimated. The daily rainfall amount is split uniformly between the as-wet-defined 8 hour-intervals. Based on the possible placements, the resulting cascade generator for the first disaggregation step is shown in Eq. 4 and substitutes Eq. 2.

$$W_1, W_2, W_3 = \begin{cases} 1, 0 \text{ and } 0 & \text{with } P(1/0/0) \\ 0, 1 \text{ and } 0 & \text{with } P(0/1/0) \\ 0, 0 \text{ and } 1 & \text{with } P(0/0/1) \\ \frac{1}{2}, \frac{1}{2} \text{ and } 0 & \text{with } P(\frac{1}{2}/\frac{1}{2}/0) \\ \frac{1}{2}, 0 \text{ and } \frac{1}{2} & \text{with } P(\frac{1}{2}/0/\frac{1}{2}) \\ 0, \frac{1}{2} \text{ and } \frac{1}{2} & \text{with } P(0/\frac{1}{2}/\frac{1}{2}) \\ \frac{1}{3}, \frac{1}{3} \text{ and } \frac{1}{3} & \text{with } P(\frac{1}{3}/\frac{1}{3}/\frac{1}{3}) \end{cases} \tag{4}$$

### 3.1.3 Introduction of a preceding cascade model (Method C)

In the modification called preceding cascade model, the position-dependency for the whole disaggregation process is extended. Hence, the modifications for Method C affect all disaggregation steps. Since only method C is referred to as preceding cascade model, method A and B can be considered as non-preceding cascade models. Besides the modified position classes definition, the disaggregation process remains identical to Method A.

An example of the position-dependency extension is illustrated in Fig. 3 (bottom) and will be used for explanation. The indices $f$ and $g$ of each time step $Z_{f,g}$ represent an index for each time step ($f$=1, 2, .., $n$ with $n$=length of the time series) and each disaggregation level ($g$=1, 2, …, 7), respectively.

For a time step $Z$ ($Z_{2,1}$) the wetness state of the time steps before ($Z_{1,1}$) and afterwards ($Z_{3,1}$) of the same disaggregation level are taken into account for the identification of the position so far (so-called "non-preceding" in Fig. 3). Hence, the type of splitting (1/0, 0/1 or x/(1-x)) is chosen independently from the wetness state of two already disaggregated time steps ($Z_{1,2}$ and $Z_{2,2}$) in the next disaggregation level. For the position definition in the preceding cascade model, the information about the wetness state of the two finer, already disaggregated time steps ($Z_{1,2}$ and $Z_{2,2}$) and the following, coarse time step ($Z_{3,1}$) is taken into account (according to Lombardo et al. (2012, 2017)).

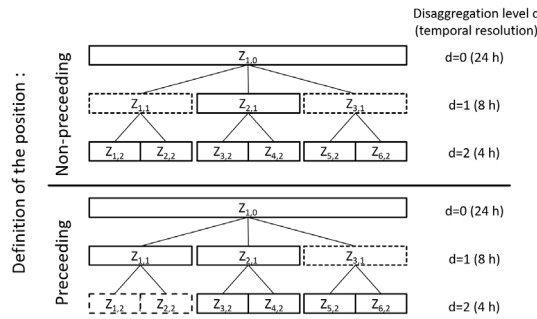

**Fig. 3: Scheme for position definition in a non-preceding (upper part) and preceding (lower part) cascade model. The dashed boxes indicate the time steps taken into account for position definition of the time step $Z_{2,1}$.**

Due to the new definition, the number of positions is extended from four in the non-preceding cascade model positions (starting, ending, enclosed, isolated) to eight in the preceding cascade model (see also Fig. 4): one starting ({0,0},1,1 with 0=dry and 1=wet and {} indicating the wetness state of the preceding, already disaggregated time steps), three enclosed ({0,1},1,1; {1,0},1,1 and {1,1},1,1), three ending {0,1},1,0; {1,0},1,0 and {1,0},1,0) and one isolated position ({0,0},1,0) for $b$=2.

## 3.1.4 Comparison of cascade model variants

Due to the new introduced position definitions a comparative overview of the different position classes (Fig. 4) and the resulting number of cascade model parameters (Table 2) is provided. For the sake of comparability, the empirical distribution function used for the x/(1-x)-splitting for $b$=2 is considered simplified as an additional parameter, since the complexity of the disaggregation method is higher with $f(x)$ than without. Nevertheless, it remains an empirical distribution function and is not a single parameter value.

From Table 2 it is visible, that the introduction of a position-dependency for $b$ =3 for cascade model B and the refinement of the position definition for cascade model C leads to an increase in cascade model parameters. Especially for method C the number of possible distributions of rainfall amounts in the already disaggregated time step before the current time step to disaggregate (see Fig. 4) leads to a strong increase of model parameters. Since all model parameters are estimated directly from observations (Carsteanu and Foufoula-Georgiou, 1996) as mentioned before, no parameter calibration is required and there is no problem with equifinality.

However, especially for the $b$ =3-splitting for the upper volume class in method C, the number of parameters is critical. Since only days with rainfall amounts higher than the $q_{0.998}$ quantile are taken into account, only a few days exist for the parameter estimation if the observed time series for parameter estimation is not long enough. This will lead to probabilities with $P$=0 for several splittings. While for some splittings $P$=0 seems reasonable from a physical interpretation (e.g. P(½/0/½)=0 is reasonable, since the highest daily rainfall amounts have no dry spell in between with a minimum of 8 h in the observed data set), for other probabilities this can result from the too small population for parameter estimation. To ensure the applicability of method C in practice, a cross-validation is carried out (see Appendix 1).

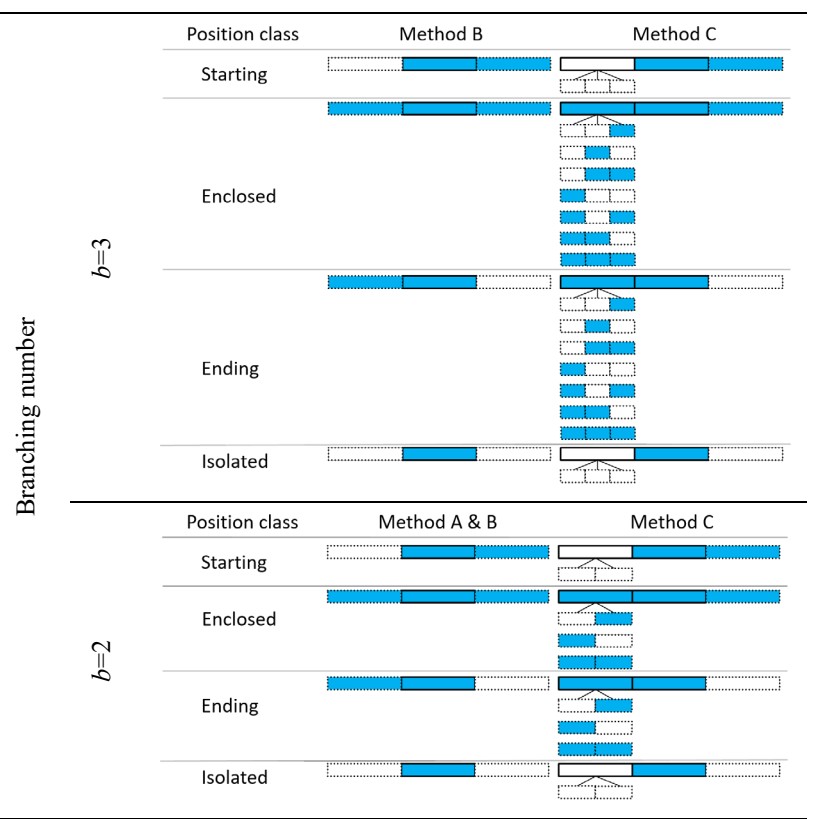

**Fig. 4: Comparison of position classes definition for methods A, B and C. For method A, no position classes differentiation is applied for *b*=3. The dashed boxes indicate the time steps, which are analysed regarding their wetness state for the definition of the position class. Blue boxes indicate a wet time step, white boxes a dry time step.**

## 5   3.2 Resampling algorithm

A different way to increase the autocorrelation is a resampling of the disaggregated time series. In a resampling process, two elements (here: relative diurnal cycles of the disaggregated time series) are swapped to improve an objective function (here: minimizing the deviation of the autocorrelation function of the disaggregated time series from the observed time series). In a relative diurnal cycle the diurnal cycle with absolute rainfall amounts per time step is transformed by dividing the single

rainfall amounts by the total rainfall amount of that day. With the simulated annealing algorithm as a resampling method it is possible to find the global optimum of an objective function. Simulated annealing has been used before for the optimization of the autocorrelation of rainfall time series by Bárdossy (1998). However, these time series were simulated by a different rainfall generator. The resampling algorithm in this study is applied with the aim to improve the autocorrelation function of the disaggregated time series under the following restrictions:

15           a) The structure of position and volume classes in the disaggregated time series generated by the cascade model
              should be conserved.

              b) The daily rainfall amount should be conserved exactly.

For a) the restriction is fulfilled by allowing only swaps of time series elements among subsets of the same position and volume class. Restriction b) is fulfilled by swapping only relative diurnal cycles as time series elements, which does not

affect the daily rainfall amount.

The objective function of the simulated annealing algorithm applied in this study is:

$$O_{auto} = \sum_{i=1}^{NoLags} (r(i) - r(i)^*)^2 \tag{5}$$

The quantities indicated by * are the prescribed values for each lag from observed time series for each station, the other quantities are the current values. *NoLags* represents the number of lags analysed for the representation of the autocorrelation

function. The number and selection of lags was carried out in a sensitivity analysis before by Föt (2015), resulting in 72 lags, whereby every second lag from lag 1 (5 min) to lag 144 (12 h) is taken into account (lag 1, lag 3,…, lag 143). After 12 hours, the values of the autocorrelation of the observed time series show an asymptotic behaviour indicating a very low process memory. As proven by Müller and Haberlandt (2018), the resampling does not affect the scaling behaviour of the disaggregated time series (see Sect. 3.4), because the total rainfall amount as well as the number of wet time steps are kept.

In a prior study (Legler, 2017) the effect of the resampling algorithm on the extreme rainfall values was analysed. Without taking the extreme values into account explicitly in the objective function, the resampling leads to a decrease of the extreme rainfall values. Since the extreme rainfall values are represented well after the disaggregation (Fig. 10), they would be

underestimated after the resampling. Since the occurrence date and the magnitude of the extreme rainfall values differs among the investigated durations, for their identification an event-independent, general scheme has to be applied in order to take them into account in the objective function. The applied scheme in this investigation is illustrated in Fig. 5. A threshold intensity $I_{tr}$ is chosen for the whole time series, whereby the first and the last time step of each day exceeding $I_{tr}$ determine the event and its duration $D_{event}$. During the resampling, swaps are only allowed if the following restrictions R are fulfilled:

> RI) The total rainfall amount of the extreme event must not decrease.

> RII) The number of dry time steps inside the extreme event must not decrease.

If $I_{tr}$ is chosen too high, extreme rainfall events of higher durations and most often lower intensities are not considered. If $I_{tr}$ is chosen too low, too many rainfall events are considered as extreme events, which leads to a rejection of too many swaps during the resampling and hence only minor improvements of the autocorrelation function.

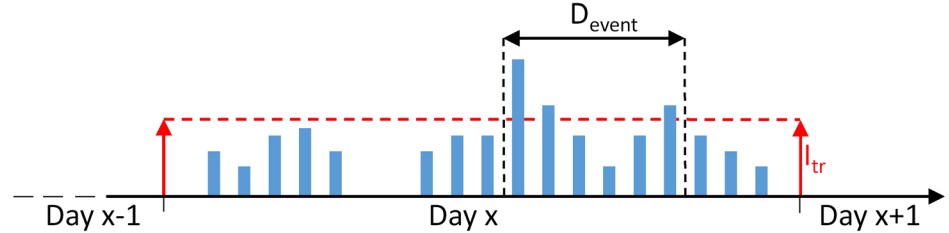

**Fig. 5: Scheme for extreme rainfall event definition**

Hence, the choice of an appropriate $I_{tr}$ is essential for a successful resampling. Since it shall be possible to estimate $I_{tr}$ a priori without a calibration of this parameter, a transferable method was required. Müller and Haberlandt (2018) identified the existence of dry time steps inside extreme events of the disaggregated rainfall time series, while the observed extreme events consisted only of wet time steps. Since an event-based simulation of extreme rainfall events with $D=30$ min in an urban hydrological model led to satisfying results regarding flooding volume and combined sewer overflow volume in Müller and Haberlandt (2018), for the identification rainfall extreme events with $D=60$ min are considered in this study. For the observed time series of all stations, the average intensity of the extreme rainfall event with $D=60$ min and the empirical return period closest to $T_n=1.5$ years was calculated. It is assumed that the results regarding under- or overestimation for the return period $T_n=1.5$ years are representative for typical return periods for dimensioning purposes in urban hydrology ($T_n=\{1, 2, 5, 10 \text{ years}\}$, DWA-A 531 (2012)). The resulting average intensity $I_{tr}=1.0$ mm of the aforementioned extreme rainfall events is applied throughout the resampling and represents the 0.99 quantile for 25 % of all stations (ranging from ~0.987 to ~0.994 between all stations). Similar thresholds have been applied before in literature for tracking of convective cells in radar images ($I_{tr}>0.7$ mm, Handwerker (2002)).

Since the number of diurnal cycles is limited in the disaggregated time series, the degree of improvement is limited as well, which can be a serious problem, if only short daily rainfall time series are available. A possible solution is to enable the

swapping of relative diurnal cycles between different realisations of disaggregated time series to increase the number of possible swap elements by additional realisations. Here the time series length was found to be sufficient and the resampling was carried out for each realisation separately.

The simulated annealing is carried out singular for each station as follows:

1. For each wet day the relative diurnal cycles are constructed. Subsets $y$ for each applied position-volume-class combination are created with $y=1,...,S$.

2. A subset $y$ is identified randomly. The probability for being identified is based on the number of elements $m$ in the subset:
$$P_{y,i=}\frac{m_i}{\Sigma_{i=1}^{S} m_i} \qquad (6)$$

3. Two days are drawn randomly from the identified subset, their diurnal cycles are swapped. If R I and R II are not fulfilled, the swap is retracted and the algorithm proceeds with step 2.

4. $O_{auto}$ (Eq. (5)) is updated.

5. The updated value $O_{auto,new}$ is compared with the former value $O_{auto,old}$. If $O_{auto,new} < O_{auto,old}$ the objective function value has improved and the swap is accepted.

6. If $O_{auto,new} \geq O_{auto,old}$ the swap is accepted with the probability $\pi$:
$$\pi = \exp(\frac{O_{auto,old}-O_{auto,new}}{T_a}), \qquad (7)$$
where $T_a$ is the annealing temperature that controls the acceptance of bad swaps. Local optima can be left by the acceptance of bad swaps and the global optimum can be found. The decrease of the annealing temperature during the resampling (see step 8) leads to a lower probability for accepting non-improving swaps, enabling the identification of the global optimum.

7. Steps 2-6 are repeated $K$ times.

8. Reduction of the annealing temperature:
$$T_a = T_a \times dt \qquad \text{with } 0 < dt < 1 \qquad (8)$$
After reducing the temperature, the algorithm proceeds to step 2.

9. Steps 7 and 8 are repeated until the algorithm converges, expressed by $M$ swaps which do not lead to an improvement of $O_{auto}$ higher than a certain threshold $thr_{O,auto}$.

The following setup was chosen for the resampling: $T_{a,start}=1*10^{-4}$, $dt=0.99$, $K=500$, $M=200$ and $thr_{O,auto}=1*10^{-9}$.

The different variants of the cascade model can be combined with the resampling approach for the improvement of the autocorrelation. A summary of the combinations and their abbreviations used throughout the manuscript are presented in Fig. 2.

## 3.4 Validation of the results

For the evaluation of the disaggregation process, the disaggregated rainfall time series are analysed regarding their scaling moments, different event-based and continuous rainfall characteristics and their extreme values. Since the cascade model is based on the scaling theory, it is proven if the disaggregated time series show the same scaling behaviour as the observations. Scaling behaviour is analysed with the relation:

$$M_q = \lambda^{K(q)} \tag{9}$$

with moments $M$, moments order $q$, the moments scaling exponent $K(q)$ and the scale ratio $\lambda$. The scale ratio represents a dimensionless ratio of two temporal resolutions of one time series. Dry time steps are neglected as well as time steps with rainfall intensities <0.1 mm to reduce the impact of too small rainfall intensities and the transformation applied after the disaggregation to reach a final temporal resolution of 5 min. It is common to analyse the scaling behaviour with log-log-plots of $M_q$ and $\lambda$ (Over and Gupta, 1994, Svensson et al., 1996, Burlando and Rosso, 1996, Serinaldi, 2010, Müller and Haberlandt, 2018). Moments are estimated as probability-weighted moments (Yu et al., 2014, Ding et al., 2015) due to their robustness against large rainfall intensities (Kumar et al., 1994, Hosking and Wallis, 1997). According to Kumar et al. (2014) and Lombardo et al. (2014) only the first three moments 1≤q≤3 are analysed.

For an event-based evaluation, first the rainfall events are identified and then the characteristics of these events are determined. Event-based rainfall characteristics include wet and dry spell duration as well as wet spell amount. An event is hereby defined as a wet period enclosed by at least one 5 min time step without rainfall before and after the wet period.

For a continuous-based evaluation, the whole time series is considered, without differentiation into single events. As continuous time series characteristics, the average intensity, the fraction of dry intervals and the autocorrelation are analysed.

For the extreme rainfall event analyses, the event definition differs to ensure the independence of the extreme events. The definition depends on the extreme event duration under investigation. For extreme event durations shorter than 4 hours, a minimum of four dry hours before and after the event ensure the independence of the event (Schilling, 1984). For longer extreme event durations, the same duration as under investigation has to be dry before and after the event. To increase the population of extreme events, partial duration series are extracted from the time series instead of annual maxima. Partial duration series are similar to the peak-over-threshold approach, whereby the threshold is defined in order to select 3 extreme rainfall events on average per year. Since the lengths of the time series of the analysed stations differ, theoretical distribution functions are fitted to enable comparisons for the same return periods among the stations. Following the DWA-A 531 (2012), which is a technical standard in Germany, an exponential distribution is fitted to the median of the realisations for each station.

To enable comparisons of rainfall characteristics at the same location, observed 5 minute time series (*Obs*) are aggregated to daily values and then disaggregated back to 5 minute time series (*Dis*). A split-sampling into calibration and validation

period was not carried out to keep the time series as long as possible for the parameter estimation (see also the discussion in Section 3.1.4).

The disaggregation is a random process. Depending on the choice of the random number generator initialization different realisations are generated. This uncertainty is taken into account by performing 30 realisations for each station. By 30 realisations the random behaviour of the disaggregation process is fairly well covered. The mean and the range of the event-based and continuous rainfall characteristics were plotted against the number of realisations used for their estimation, and an asymptotic behaviour was identified with increasing numbers of realisations after 30 realisations.

For the validation the relative error $rE$ and relative absolute error $rAE$ are calculated to quantify the direction and the amount of the deviation of the rainfall characteristic $RC$ with $i$ as control variable of all realisations $n$:

$$rE = \frac{1}{n} \times \sum_{i=1}^{n} \frac{RC_{Dis,i} - RC_{Obs,i}}{RC_{Obs,i}}, \tag{10}$$

$$rAE = \frac{1}{n} \times \sum_{i=1}^{n} \frac{|RC_{Dis,i} - RC_{Obs,i}|}{RC_{Obs,i}}, \tag{11}$$

## 4. Results

### 4.1 Modifications to cascade model

For an improved representation of the autocorrelation function two modifications of the multiplicative cascade models after Müller and Haberlandt (2018) have been analysed, namely method B and method C. For method B, the order of wet and dry 8 hour intervals is not assigned randomly as in Müller and Haberlandt (2015). Probabilities for each combination of wet and dry 8 hour-intervals are estimated, with a differentiation according to the position of the daily time step in the time series (starting, enclosed, ending or isolated).

The resulting probabilities are shown in Table 3, 4 and 5 (columns with position-dependent entries) in comparison to the position-independent probabilities estimated for method A (first column in each table). For starting positions, splittings with wet 8 hour-intervals at the end of a day have the highest probabilities (for both one and two wet intervals). For ending positions, a vice versa relationship can be identified with highest probabilities for wet 8 hour-intervals at the beginning of a day. For enclosed positions, probabilities for a wet 8 hour-interval at the beginning or ending of the day, so with a temporal connection to another wet day, are higher, if one interval is wet. All of these findings are similar to the findings from Olsson (1998) and Güntner et al (2001) for a splitting with $b=2$. Also, independent from the position, it can be identified that probabilities for two connected wet intervals (1-1-0 and 0-1-1) are higher than the combination with an enclosed dry time step (1-0-1). The probability, that three intervals are wet, is the highest for enclosed position and the lowest for isolated position.

The scaling behaviour of observed and disaggregated time series is shown in Fig. 6 for station Cuxhaven. The results are similar among the stations and the analysed moments. The disaggregated time series show an identical scaling behaviour

down to a temporal resolution of 120 min. For finer resolutions slight overestimations occur for method A and B and even slighter for method C. These deviations are presumably caused by fragments of the linear transformation applied after the disaggregation to achieve a final temporal resolution of 5 min. Overall, with a mean deviation of all three methods A-C from the observations of 0.08 for $q=1$ and a temporal resolution of 5 min the scaling behaviour is represented well. T

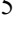

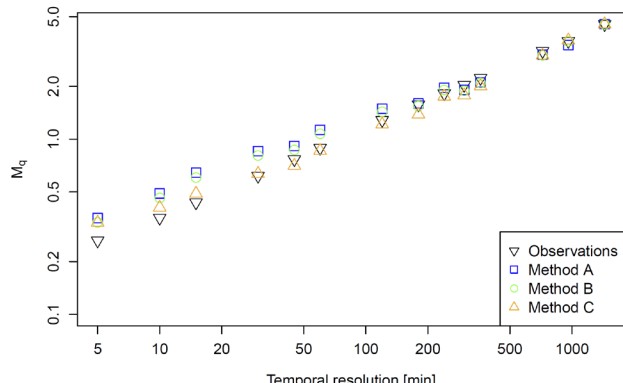

**Fig. 6: Scaling behaviour of observed and disaggregated time series for station Cuxhaven for $q=1$. Each value represents the mean of 30 realisations.**

The rainfall characteristics of the disaggregated time series are shown in Fig. 7 in comparison to observations. A quantitative analysis of the deviations is provided in Table 6 with relative $rE$ and absolute errors $rAE$ (see Eq. 10 and 11) for the mean values of rainfall characteristics. Method A represents the original model proposed by Müller and Haberlandt (2018) and will be referred to as reference for the evaluation. Since the MMD approach is investigated for the first time, its influence on the rainfall characteristics of the disaggregated time series is analysed. If the too small rainfall intensities lower than the instrumental resolution of the measurement device are kept after the disaggregation (hence, MMD is not applied), the results are referred to as 'A/B/C-standard'. If the too small rainfall intensities are eliminated by the MMD approach, the results are referred to as 'A/B/C-MMD'.

While for method A-standard a slight overestimation for the average intensity is identified ($rE=11$ %), for wet spell duration (-3 %) and amount (8 %), dry spell duration (8 %), fraction of dry intervals (1 %) and lag-1 autocorrelation (-4 %) a good representation is achieved.

With the introduction of a position-dependence in the disaggregation step from daily values to 8 h-values in method B-standard an improvement of all rainfall characteristics can be achieved. The improvements of the wet and dry spell durations are direct consequences of the better representation of the wetness state of 8 h-intervals as is indicated by the parameter values in Table 3, 4 and 5. For the average intensity with $rE=3$ % a major improvement from an overestimation of $rE=11$ % (A-standard) is identified.

For method C-standard, a worsening of the majority of rainfall characteristics is identified. Wet spell duration is overestimated with $rE$=399 %. This is caused by the high probability for a $x/(1-x)$-splitting for enclosed boxes in the preceding cascade model, which decreases the probability for splitting one event into two events by the generation of dry time steps. This leads to a high number of wet time steps (underestimation of fraction of dry time steps $rE$=-15 %) and

5 consequently to an underestimation of average rainfall intensities ($rE$=-71 %) due to the exact mass conservation of the cascade model.

Too small intensities can be avoided by the MMD approach. For method A and B, the exclusion of small rainfall intensities leads to a worsening of rainfall characteristics (Fig. 8, Table 6). This indicates, that the before mentioned good representation of rainfall characteristics by method A-standard and B-standard is biased by wet time steps with rainfall amounts lower than

10 the observed minimums (depending on the instrumental resolution of the measurement device). Since time steps with these rainfall intensities are negligible from a hydrological point of view, the line of MMD in Fig. 7 provide a more useful insight into the disaggregated data.

The overestimation of the average rainfall intensity by methods A and B increased to $rE_{MMD}$=40 % and 32 %, respectively, while the underestimation by method C is reduced to $rE_{MMD}$=-35 %. An improvement for wet spell duration is also identified

15 ($rE_{MMD}$=-16 %). Although the fraction of dry intervals improved with MMD ($rE$=-3 %), a worsening of the dry spell duration is identified ($rE_{MMD}$=-47 %), indicating higher fraction of short dry spells inside former events on a coarser time scale.

Method A and B result in similar values for wet spell duration as for method C for MMD. For wet spell amount and duration, average intensity, dry spell duration and fraction of dry intervals a decrease in performance is identified by MMD in comparison to the standard approach.

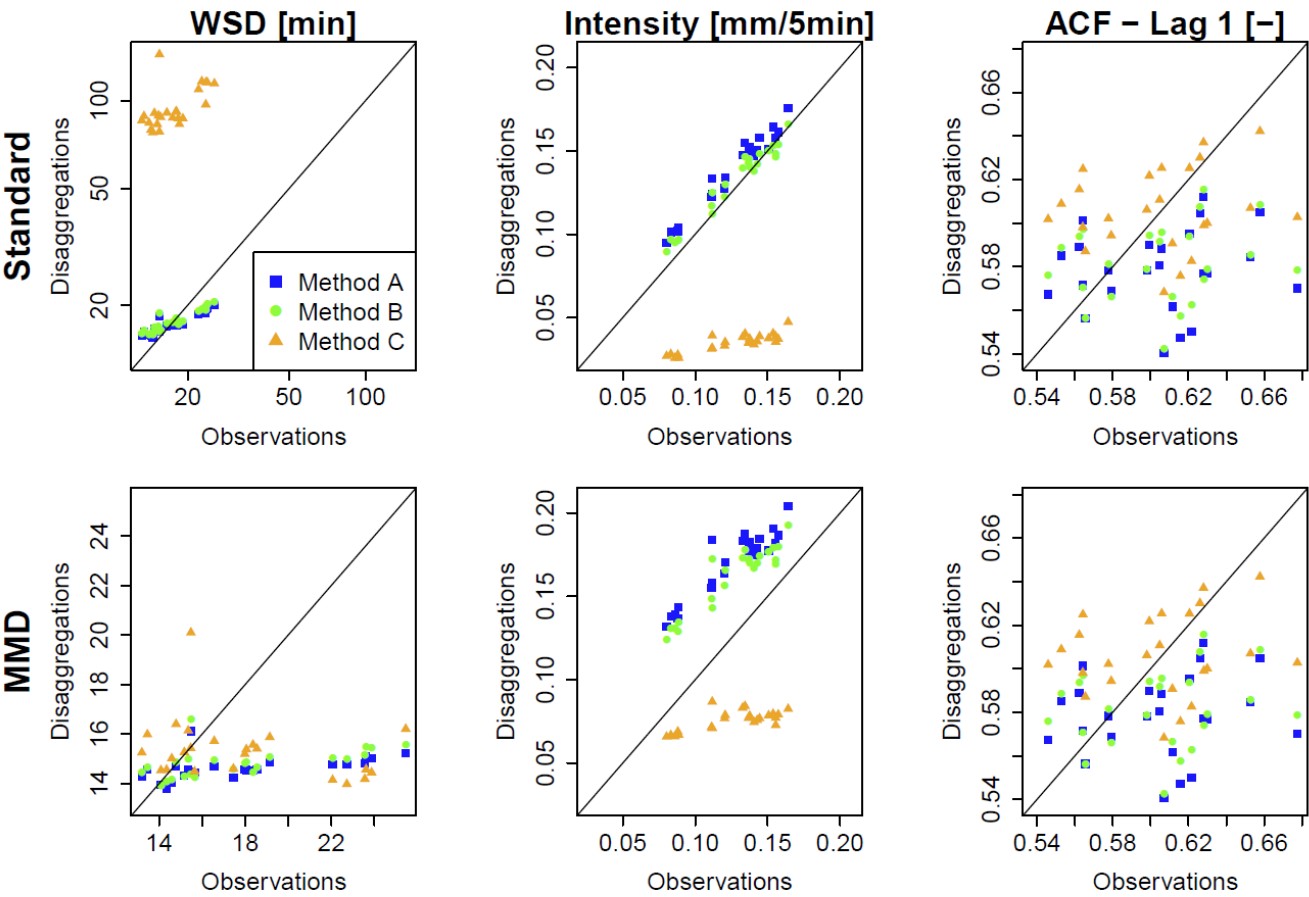

**Fig. 7: Rainfall characteristics of observed and disaggregated time series as x-y plots for all 24 stations (WSD=wet spell duration, ACF=autocorrelation function; note the different scales for wet spell duration)**

5    Since the focus of this study is the improvement of the autocorrelation, the impact of MMD on method A, B and C is investigated as well. From a visual inspection of the lag-1 autocorrelation in Fig. 7, a systematic underestimation as mentioned in the Sect. 1 is not visible, since for some stations even overestimations occur. However, a comparison between observations and disaggregated time series resulting from different methods until lag 144 (representing a time shift of 720 min=12 h) shows differences and a clear underestimation by the disaggregation for station Braunlage (Fig. 8). For other

10   stations the relationship is similar, although for some the differences between method A and B are smaller. In Fig. 9 the relative error between the median of the autocorrelation function of all 30 realisations for each method and the observed time series is shown for all stations regarding lag 1 (5 min), 6 (30 min) and 36 (180 min). Independently of the applied methods, the deviation is increasing from lag 1 to lag 6, while for lag 36 the deviation has decreased. Also, the range of deviations is decreasing for an increasing number of lags. This is visually confirmed by the results for station Braunlage (Fig. 8), where

the autocorrelation of the disaggregated time series decreases strongly with the first lags, while it decreases much smoother for the observed time series. The choice of the disaggregation method (method A, B or C) has a higher impact on the resulting autocorrelation than the choice of treatment of the too small rainfall intensities (Standard or MMD). In fact, the MMD approach has only a slight effect on the autocorrelation function values. The smallest deviations of the autocorrelation function are achieved with method C, independent from the treatment of the too small rainfall intensities.

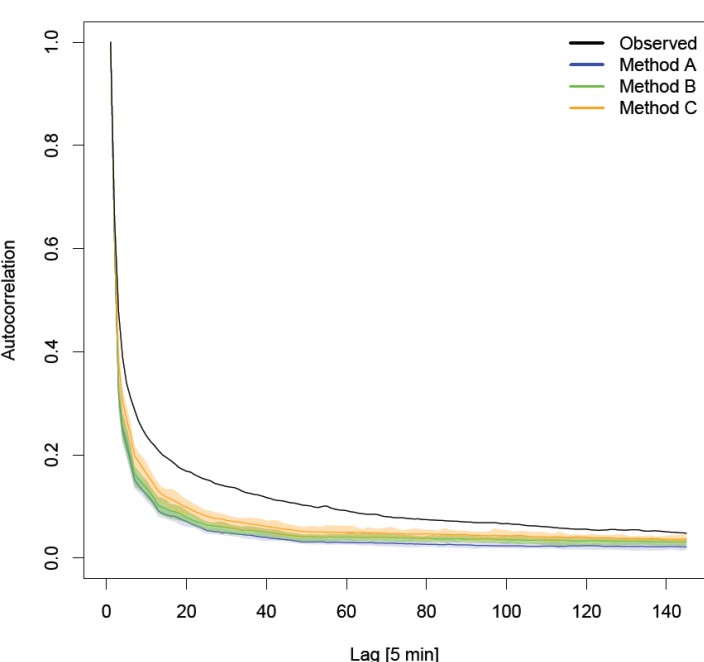

**Fig. 8: Autocorrelation of observed and disaggregated time series using the standard approach for each method with no modification regarding too small rainfall intensities. The range for each method results from 30 realisations, the solid line represents the median.**

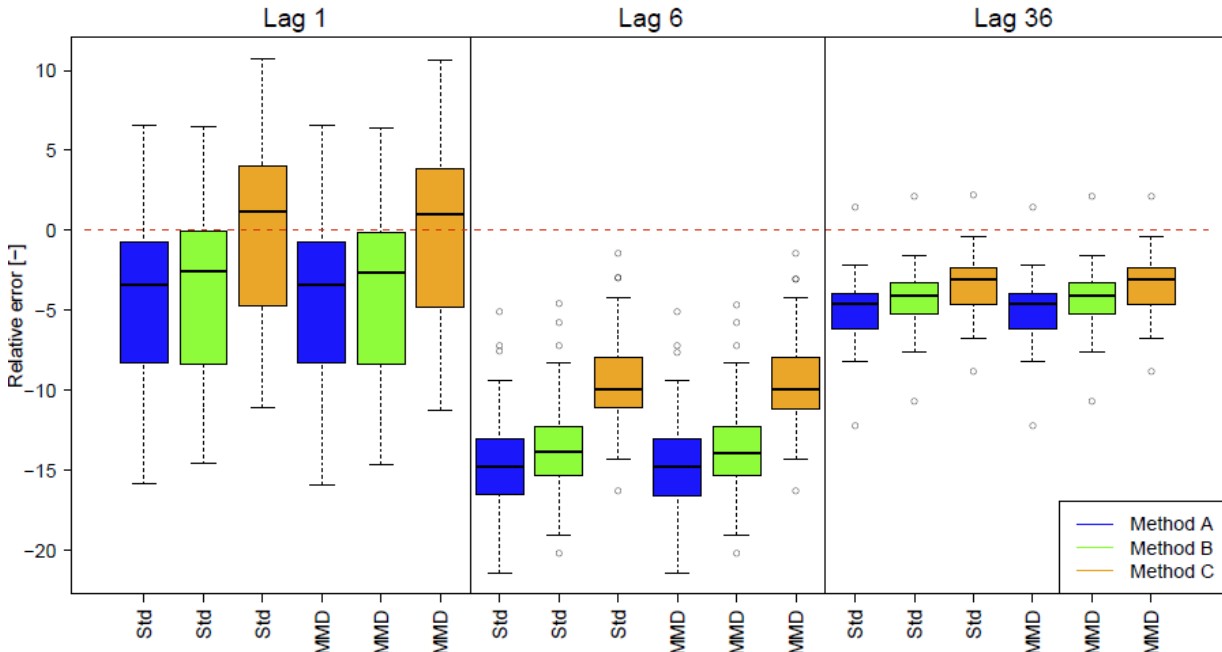

**Fig. 9: Deviations of autocorrelation from disaggregated to observed time series as relative error for lags 1, 6 and 36. The red dashed line indicates a $rE=0$ (Std is used as abbreviation for Standard)**

The results of the extreme rainfall value analysis are illustrated in Fig. 10 for two durations $D$ (5 minutes and 1 hour) and two return periods $T$ (1 year and 5 years). For the extreme events, only between methods A, B and C is differentiated. The modifications regarding the minimum rainfall intensity are not taken into account since they do not affect the rainfall extreme events.

For a return period of $T=1$ year extreme rainfall values are slightly overestimated by less than $rE=10$ % for the half of all stations and less than approximately $rE=20$ % for 75 % of all stations for both analysed durations, independent of the applied modification of the cascade model. For $T=5$ years, the range of results is increasing, leading to a worse representation in comparison to $T=1$ year. While for $D=5$ min a slight overestimation of approx. $rE=10$ % for half of all stations can still be identified, for $D=1$ hour an underestimation of $rE=50$ % is identified for half of all stations. However, increasing deviations with increasing return periods can be expected, since for a few of the time series with lengths of only 9 years the return period is limited to $T=3$ years (1/3 of time series length) to ensure plausibility from a hydrological point of view.

Nevertheless, it should be noted that over all return periods and durations, method C lead to the smallest range of relative errors over all stations in combination with the best fit to the distribution of the observed extreme rainfall values. The cascade model parameter transferability in space was confirmed by a cross-validation for method C (see Appendix A).

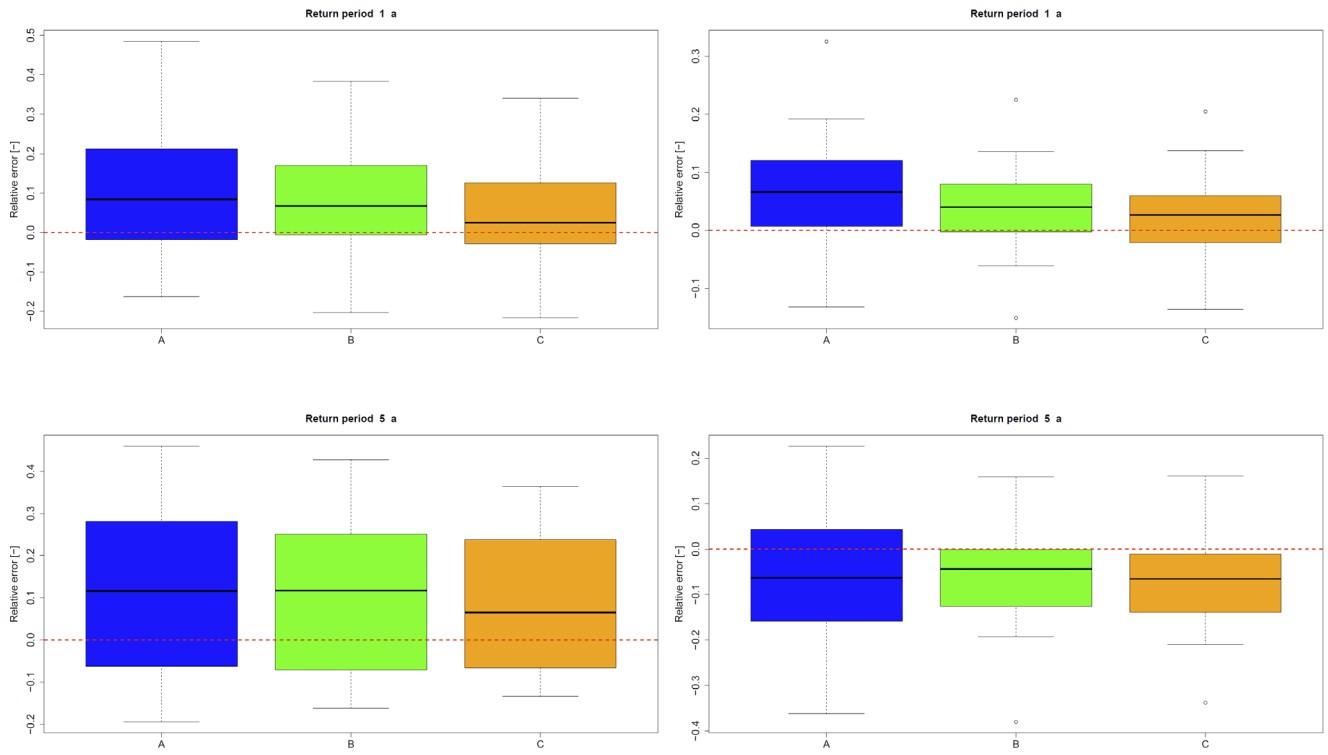

**Fig. 10: Mean relative errors of extreme values of the disaggregated time series for all stations (the dashed line represents an error of 0). Results are shown for durations of 5 min (left) and 1 h (right) and for return periods of 1 year (top) and 5 years (bottom).**

## 4.2 Resampling results

For the resampling, only disaggregated time series modified by the MMD approach are used due to their slight better representation of the autocorrelation. The autocorrelation of the disaggregated time series before and after the resampling are shown in Fig. 11 for lag 1, lag 6 and lag 36. A general increase of the autocorrelation along with smaller deviations for the median of all stations compared to before the resampling can be identified for all three methods A, B and C. Only for the lag 1-autocorrelation of the rainfall time series disaggregated with method C does the resampling lead to a worsening regarding the median value. However, the range of the lag 1-autocorrelation results is reduced, indicating that the under- and overestimations were reduced by the resampling approach.

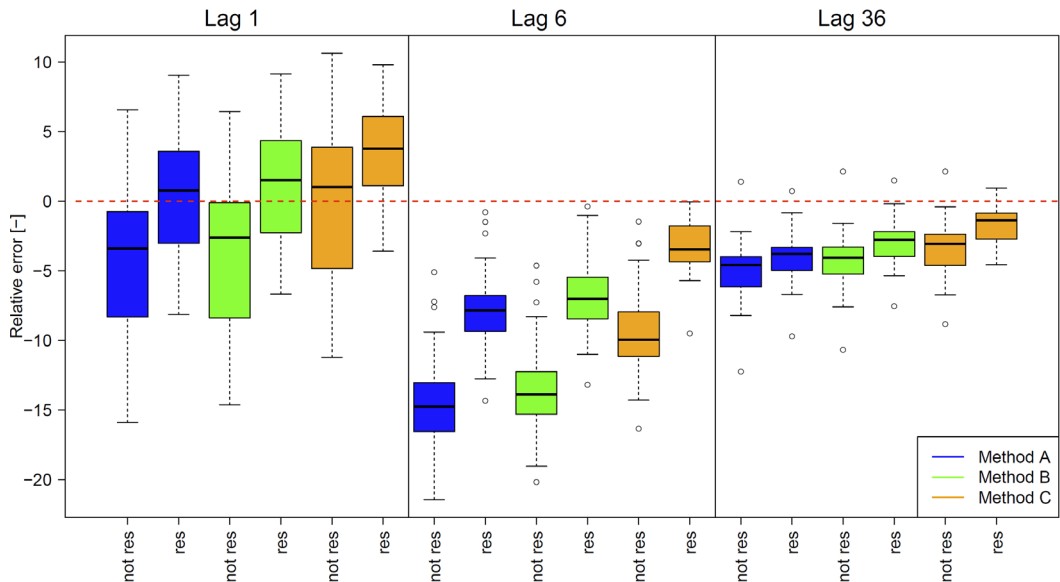

**Fig. 11: Deviations of autocorrelation from disaggregated to observed time series before and after the resampling as relative error for lags 1, 6 and 36. All results are based on the MMD approach. The red dashed line indicates a *rE*=0, results for the resampled time series are labelled with 'res'.**

As mentioned before, the improvement of the autocorrelation depends on the chosen threshold for extreme rainfall value definition, $I_{tr}$. An increase of $I_{tr}$ leads to a decrease of the number of rejected swaps during the resampling, since less time steps are involved in the extreme value analysis. An unrealistically high value of $I_{tr}$ (identical with leaving out both restrictions RI and RII regarding extreme rainfall values) leads to almost perfect fits for lag 1 and lag 36 ($|rE|<1$ % for the majority of the stations), and for lag 6 deviations up to $|rE|<3.5$ % occur (not shown here). However, the extreme rainfall values are underestimated strongly if $I_{tr}$ is chosen too high.

Hence, both restrictions RI and RII are applied during the resampling by the choice of $I_{tr}$=1 mm. In Fig. 12, the extreme rainfall event series for station Osnabrück is shown for *D*=5min. Although the extreme event series changed slightly after the resampling, the overall extreme series characteristics regarding range, under- and overestimation in comparison to the observations remain the same for all return periods.

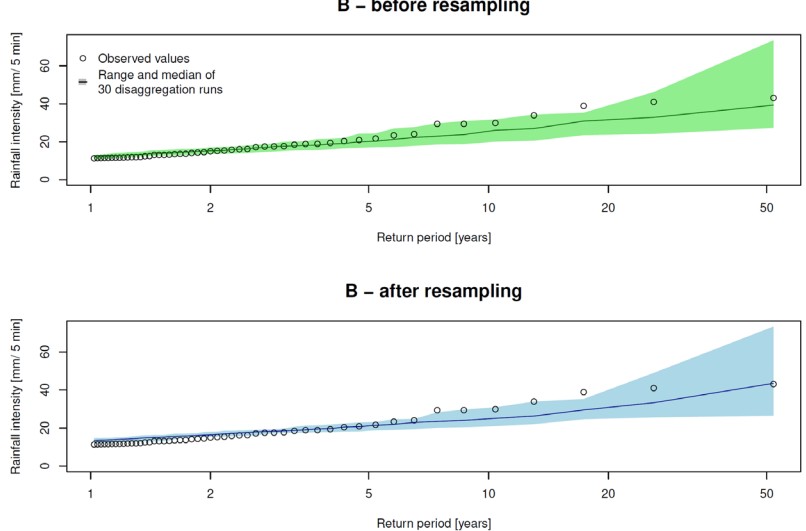

**Fig. 12 Extreme rainfall values for *D*=5 min for station Osnabrück based on 30 disaggregation realisations with method B before (upper part) and after (lower part) the resampling**

For extreme rainfall events with longer durations (*D*={15 min, 1 h, 2 h}) the impact of the resampling is quantified in Table 7. The impact of the resampling depends on the analysed duration of the extreme rainfall events. While for *D*=15 min the median of *rE* has decreased after the resampling with smaller |*rE*| for the smaller return periods ($T_n$={1, 2 years}) and higher |*rE*| for $T_n$=10 years, for *D*=2 h the median of *rE* has increased after the resampling with higher |*rE*| for the smaller return periods and smaller |*rE*| for $T_n$=10 years.

Since these findings are independent from the disaggregation method, the differences are caused only by the resampling. Extreme rainfall events with *D*=15 min represent convective events with only a few wet time steps preceding and succeeded by dry time steps. Due to the short event duration, the possibility of dry time steps in between is small and RI is the active restriction, which requires the total rainfall amount not to decrease, resulting in an increase. Extreme rainfall events with *D*=2 h originate from long-lasting, stratiform events with a high fraction of wet time steps in the current day. Since this fraction of wet time steps can also be found in the disaggregated time series, the rainfall amount will be distributed on more wet time steps to fulfil the active restriction RII to increase the autocorrelation.

However, the majority of *rE* values presented in Table 7 is smaller than 10 %, which indicates a good representation of the extreme rainfall events in the disaggregated time series in general, independent of the application of the resampling algorithm.

## 5. Discussion

### 5.1 Impact of the cascade model modifications

The two new introduced micro-canonical cascade model variants methods B and C differ regarding their way of how the disaggregation process depends on a position of a wet time step in a rainfall time series and how it is defined. Both, the position dependency in the first disaggregation step with $b=3$ for method B and the position definition after Lombardo et al. (2012, 2017) for method C, are based on additional parameters of the disaggregation model (see Table 2). However, all new parameters are process-based and physically interpretable, since they describe the rainfall dependency of past time steps. Hence, an improvement of the autocorrelation, which describes the process memory, was expected. While method B differs from method A only in the first disaggregation step, a smaller improvement of the autocorrelation can be identified, while method C differs in every disaggregation step and thus a higher improvement is identified.

As all parameters of the micro-canonical cascade model, including the newly introduced parameters can be estimated from the aggregation of observed high-resolution rainfall time series (Carsteanu and Foufoula-Georgiou, 1996), no additional calibration has to be carried out. To reduce the increase of number of parameters by method B and C, several possibilities exist. Olsson (1998), Güntner et al. (2001) and Müller and Haberlandt (2018) identified similarities between cascade model parameters of different position classes which can be used for simplification. These similarities are e.g. P(0/1) for starting and P(1/0) for ending positions (and vice versa) as well as P(0/1) and P(1/0) for both, enclosed positions and isolated positions. Another possibility is to apply a semi-bounded cascade model instead of a bounded cascade model. While in a bounded cascade model for each step of the disaggregation process the corresponding parameter set is used (as it is done in this study), in a semi-bounded cascade model the same parameter set could be applied over a range of disaggregation levels as long as a mono-fractal scaling behaviour can be assumed. Based on Veneziano et al. (2006) typical ranges for mono-fractal behaviour are from daily to hourly resolution and from hourly to 5 minute resolution.

It should also be noted that the analyses of only the lag-1 autocorrelation is not sufficient, since it provides a limited insight into the process memory. Here, for some stations an overestimation of the lag-1 autocorrelation was identified, but underestimations for lag-6 and lag-36. Hence, a multi-lag analyses is recommended for further studies.

Especially for method C, the general problem of the micro-canonical cascade model of generating time steps with too small rainfall intensities (Molnar & Burlando, 2005, Müller and Haberlandt, 2018) worsened. The MMD approach is introduced to solve this issue. MMD simulates the behaviour of a measurement device (minimum rainfall amount required to cause a registration) after the disaggregation process, eliminating too small rainfall intensities by summing them up to future time steps until the minimum rainfall amount is achieved or exceeded. The MMD approach has a smaller impact on the resulting rainfall characteristics than the choice of method A, B or C. However, the application of MMD leads to disaggregated time series with a slight better representation of the autocorrelation function. Hence, for the subsequent analyses only the MMD

approach is considered. This selection has also the additional advantagesthat the process of the rainfall measurement itself is simulated.

## 5.2 Impact of the resampling

After the disaggregation process a subsequent resampling approach as post-processing strategy was investigated to improve the autocorrelation as well. While the generated time series structure (defined by the position-volume-class combinations) is conserved during the resampling process, a special focus has to be given to the conservation of the extreme rainfall values. Without this focus, the resampling algorithm aims to swap diurnal cycles in a way to distribute high daily rainfall amounts on many wet time steps to generate less intense events, which leads to an underestimation of the extreme rainfall values. The universal definition of extreme rainfall values introduced here is required to conserve these extreme values a priori without any information about their date of occurrence or their magnitude. This definition and the connected restrictions for the resampling can be modified in multiple ways to improve the conservation. For example, the applied threshold intensity $I_{tr}$ can be based on a different or an additional (required) return period or duration. Also, a definition of $I_{tr}$ as a quantile of all wet time steps of a disaggregated time series instead of an absolute value would be helpful if there is a high variation of mean rainfall intensities among the investigated stations for an extreme event with a certain duration and return period (which was not the case in this study). However, the extreme values were conserved during the resampling process. The autocorrelation was improved for almost all lags, independent of whether method A, B or C was applied before for the disaggregation. Also, a higher improvement was achieved by the resampling than by the modifications of method B or C.

## 5.3 Study limitations

This study is focused on the methodological development of the micro-canonical cascade model and on the improvement of the disaggregated time series by a resampling approach as post-processing strategy. Hence, the study is limited in several aspects, which will be stated below.

First of all, the rainfall data set, with 24 rain gauges, is rather small. Although the study area covers different topographical region and climate classes, the resulting time series characteristics are similar and do not cover a wide range. A generalization of the results has to be proven for regions which are very different from this study area. To draw general conclusions from the point of comparative hydrology future research should include rain gauges from different climate regions and topologies.

Second, based on the similar rainfall characteristics and extreme values, the introduction of the universal extreme value definition was possible and representative for all stations in the study. If stations from different climate regions and topologies are studied as recommended before, it has to be proven i) if the introduced universal extreme value definition still has the potential to conserve the extreme rainfall values throughout the resampling process and ii) if $I_{tr}$ has to be redefined (see also the discussion in Sect. 5.2).

Third, as mentioned in the Sect. 1, Pearson's autocorrelation is a measure of linear dependency. It only captures the complete dependence structure between random variables if they are jointly Gaussian. An alternative criterion would be Spearman's rank correlation (capturing monotonic but not necessarily linear relationships). In both cases, autocorrelation as a function of lags is only meaningful in the context of second-order stationary stochastic processes (or weakly stationary processes).

Rainfall intensities are most likely not normally distributed. Also rainfall time series present a mixture of processes due to the high intermittency of rainfall amplified by the disaggregation process, changing between the two states of rainfall occurrence and non-occurrence. Still, every kind of autocorrelation measurement can provide a measure for the similarity of e.g. two time series. The Pearson's autocorrelation coefficient is widely used for autocorrelation analyses in hydrology. It is applied in this study to achieve a comparable similarity in the disaggregated time series as it is estimated from the

observations. Besides the mixture of processes and the limitations of Pearson's autocorrelation as a measure of dependence, the Hurst-phenomenon might also offer an additional perspective for the analysis at hand (see Koutsoyiannis (2009) for an introduction).

Fourth, although method C is based on a finding in Lombardo et al. (2012, 2017), the disaggregation method differs from the additive cascade model in Lombardo et al. (2012, 2017). Hence, the by Lombardo et al. identified problem of non-

stationarity of the disaggregation is not solved by the introduced cascade model variants and remains an open challenge for further studies.

Finally, a comment on the applied resampling algorithm. Simulated annealing was implemented in a computationally efficiently way suggested by Bárdossy (1998). After each swap the objective function is not completely newly calculated, rather updated only for the modified elements of the time series affected by the swap. Nevertheless, the resampling process

remains very time-demanding, depending on the chosen parameter setup. More recently published optimization algorithms are very promising regarding less computational times e.g. the quantum annealing approach (Heim et al., 2015, Crosson and Harrow, 2016), enabling the optimization of longer disaggregated rainfall time series or more realisations in the same time.

## 6. Conclusions

Three variants of the micro-canonical cascade model (method A-reference from Müller and Haberlandt (2018), B and C)

were assessed regarding their ability to represent the autocorrelation in the disaggregated, 5 minute rainfall time series, starting from daily totals. The methods differ regarding the position dependency in the first disaggregation step and the definition of a wet time step during the disaggregation process. The study was carried out for 24 stations in Lower Saxony, Germany, and results were analysed additionally for continuous and event-based characteristics as well as extreme rainfall values. The following conclusions are drawn based on the results:

1. The introduction of a position dependency in the first disaggregation step (method B) and especially the introduction of the position-dependency (method C) after Lombardo et al. (2012, 2017) lead to an improvement of the autocorrelation.

2. While method A and B lead to quite similar event-based and continuous rainfall characteristics, the results from method C differ significantly.

3. Method C leads to a high fraction of time steps with too small rainfall intensities in the disaggregated time series.

To avoid time steps with too small rainfall intensities, a process to mimicry the behaviour of a measurement device (MMD) was applied after the disaggregation process in combination with method A, B and C. For the following investigations, only method combinations with MMD were analysed, since the results indicated a slight better representation of the autocorrelation function.

After the disaggregation process the resampling algorithm Simulated Annealing was applied to improve the autocorrelation. The following conclusions are drawn:

4. The resampling leads to an improvement of the autocorrelation, independent of the applied disaggregation method or the investigated lag.

5. The improvement of the autocorrelation by the resampling was higher than by the choice of the cascade model modification.

6. The extreme rainfall values have to be considered during the resampling, otherwise they will be underestimated after the resampling process.

7. With the newly introduced universal definition the extreme rainfall events can be considered without the a priori knowledge of their occurrence and magnitude. Hence, the extreme rainfall values are represented after the resampling process as well as before.

The overall best representation of the autocorrelation was achieved by method C in combination with a resampling approach as post-processing strategy. Urban hydrological simulations would provide additional information about the impact of the different disaggregation methods and the resampling process on simulated hydrographs and flood events, but this is beyond the scope of this study.

**Acknowledgements**

The author thanks Annika Föt and Jonas Legler for their efforts in their student theses related to this manuscript. The author is also thankful for fruitful discussions with Thomas Müller about the conservation of extreme rainfall values during the resampling and with David Lun about the mathematic background of different aspects. Also Anne Fangmann and Patrick Hogan are acknowledged for their support with R-scripts and useful comments on an early draft of the manuscript, respectively. I am also thankful for the permission to use the data of the German National Weather Service (Deutscher

Wetterdienst DWD). Funding was provided for Hannes Müller-Thomy as a Research Fellowship (MU 4257/1-1) by DFG e.V., Bonn, Germany. The author acknowledges the TU Wien Library for financial support through its Open Access Funding Program. Finally, Elena Volpi and two anonymous reviewers as well as Nadav Peleg as associated editor are gratefully acknowledged for their contributions to improve this manuscript significantly.

**Competing interests**

The author declares that he has no conflict of interest.

**Code/Data availability**

The rainfall data is accessible from the Climate Data Center web portal of the German Weather Service
10 (https://cdc.dwd.de/portal/). The rainfall disaggregation program as well as the resampling program are both written in Fortran, so only executable files can be shared. However, the author is happy to share them on request.

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

**Appendix 1**

As pointed out in Section 3.1.4 method C and its position-definition increases model parameters. To ensure the model parameter transferability in space a leave-one-out cross-validation was applied. A daily time series was aggregated from an observed 5 min time series and is referred to as target station. The parameters for the disaggregation were estimated two times, i) from the high-resolution time series of the target station ($T$) used for the aggregation as well as ii) from the 5 min time series of the closest recording station referred to as neighbouring station ($N$). The average distance was $\overline{NT}$=38.9 km.

The results for the disaggregation are presented in Table A1 and Fig. A1, whereby the results for parameter estimation at the target station are identical with the results presented in Table 6 and Fig. 9. The rainfall characteristics of the disaggregated time series are very similar between both locations ($T$ and $N$) for parameter estimation with deviations of less than 5 % for $rE$ and $rAE$. Also the results for the comparison of lag 1, lag 6 and lag 36 autocorrelation are very similar. While for lag 1 the median tends to slightly overestimate the observations for parameters estimated at $T$, parameters estimated at $N$ lead to a slight underestimation. However, for lag 1 $rE$=0.01 and $rAE$=0.05 are identical. Also, the overall representation of the autocorrelation regarding range of the results and tendencies for over- and underestimation is similar between both locations for parameter estimation.

So although method C is based on a high number of model parameters, their transferability in space is confirmed by this leave-one-out cross validation for the study area.

**Table A1: Relative and absolute error of rainfall characteristics between disaggregated and observed time series in dependence of the location used for parameter estimation (mean for 24 stations)**

| | | rE [%] | | | | | | rAE [%] | | | | | |
|---|---|---|---|---|---|---|---|---|---|---|---|---|---|
| | | Wet spell duration | Average intensity | Wet spell amount | Dry spell duration | Fraction of dry intervals | Autocorrelation lag 1 | Wet spell duration | Average intensity | Wet spell amount | Dry spell duration | Fraction of dry intervals | Autocorrelation lag 1 |
| Method C | T | -16 | -35 | -45 | -47 | -3 | 1 | 16 | 35 | 45 | 47 | 3 | 5 |
| | N | -16 | -32 | -44 | -45 | -3 | 1 | 16 | 32 | 44 | 45 | 3 | 5 |

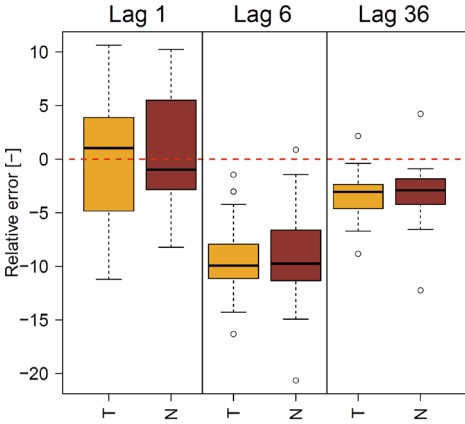

**Fig. A1: Deviations of autocorrelation from disaggregated to observed time series for method C as relative error for lags 1, 6 and 36 in dependence of the location used for parameter estimation. The red dashed line indicates a *rE*=0 (Std is used as abbreviation for Standard)**

Tab. 1: Attributes of all 24 rainfall stations, based on a temporal resolution of 5 minutes.

| ID | Name | Altitude [m.a.s.l.] | Mean annual precipitation [mm] | Fraction of wet 5 minute-intervals [%] | Average wet spell duration [min] | Average wet spell amount [mm] | Average dry spell duration [min] | Autocorrelation lag-1 [-] |
|---|---|---|---|---|---|---|---|---|
| 1 | Braunlage | 607 | 1397 | 8.1 | 15.5 | 0.51 | 175.3 | 0.66 |
| 2 | Braunschweig-Voel. | 81 | 638 | 4.4 | 15.7 | 0.43 | 336.8 | 0.62 |
| 3 | Cuxhaven | 5 | 869 | 6.2 | 19.1 | 0.51 | 291.9 | 0.61 |
| 4 | Diepholz | 39 | 690 | 4.6 | 15.2 | 0.43 | 314.8 | 0.56 |
| 5 | Emden | 0 | 825 | 5.2 | 15.5 | 0.47 | 281.2 | 0.55 |
| 6 | Freiburg/Elbe | 2 | 888 | 6.4 | 18.5 | 0.49 | 272.9 | 0.57 |
| 7 | Gardelegen | 47 | 581 | 6.2 | 22.7 | 0.40 | 340.2 | 0.63 |
| 8 | Göttingen | 167 | 631 | 4.3 | 14.1 | 0.40 | 315.3 | 0.62 |
| 9 | Hannover | 55 | 641 | 3.9 | 13.2 | 0.41 | 323.0 | 0.63 |
| 10 | Harzgerode | 404 | 612 | 7.3 | 23.9 | 0.38 | 304.3 | 0.65 |
| 11 | Jork-Moorende | 1 | 727 | 5.7 | 18.4 | 0.44 | 302.0 | 0.58 |
| 12 | Leinefelde | 356 | 942 | 8.0 | 25.5 | 0.57 | 291.1 | 0.60 |
| 13 | Lingen | 22 | 789 | 5.5 | 16.6 | 0.46 | 286.6 | 0.60 |
| 14 | Lüchow | 17 | 569 | 3.9 | 14.3 | 0.39 | 349.3 | 0.61 |
| 15 | Magdeburg | 76 | 496 | 5.5 | 22.1 | 0.38 | 373.3 | 0.62 |
| 16 | Norderney | 11 | 744 | 4.5 | 14.6 | 0.46 | 309.5 | 0.56 |
| 17 | Oldenburg | 11 | 809 | 6.4 | 18.1 | 0.43 | 263.1 | 0.63 |
| 18 | Osnabrück | 95 | 874 | 5.4 | 14.8 | 0.45 | 258.3 | 0.56 |
| 19 | Bad Salzuflen | 135 | 825 | 5.0 | 13.5 | 0.42 | 253.0 | 0.63 |
| 20 | Soltau | 76 | 804 | 5.3 | 15.4 | 0.44 | 274.1 | 0.61 |
| 21 | Uelzen | 50 | 643 | 5.5 | 17.5 | 0.39 | 300.1 | 0.58 |
| 22 | Ummendorf | 162 | 549 | 5.9 | 23.6 | 0.41 | 367.2 | 0.60 |
| 23 | Wendisch Evern | 62 | 686 | 5.8 | 18.0 | 0.40 | 290.2 | 0.55 |
| 24 | Wernigerode | 234 | 625 | 7.1 | 23.6 | 0.39 | 305.1 | 0.68 |

Tab. 2: Comparison of model parameters for methods A. B, and C in dependence of the applied branching number.

| | | Method | | |
| | | A | B | C |
|---|---|---|---|---|
| $b=3$ | Basic parameters | 3 | 7 | 7 |
| | Position classes | - | 4 | 16 |
| | Volume classes | 2 | 2 | 2 |
| | **Parameters per disaggregation step** | **6** | **56** | **224** |
| $b=2$ | Basic parameters | 4 | 4 | 4 |
| | Position classes | 4 | 4 | 8 |
| | Volume classes | 2 | 2 | 2 |
| | **Parameters per disaggregation step** | **32** | **32** | **64** |

Tab. 3: Position-dependent and –independent probabilities for one wet 8 hour interval in the uniform splitting (mean of all 24 stations for lower volume class, all values in percent [%]). The combination of '1' (wet) and '0' (dry) illustrates the order of wet and dry 8 hour intervals in a day.

| One wet interval, position-independent | One wet interval, position dependent | | | | | | | | | | | | | | | |
|---|---|---|---|---|---|---|---|---|---|---|---|---|---|---|---|---|
| | starting position | | | | enclosed position | | | | ending position | | | | isolated | | | |
| | 001 | 010 | 100 | Σ | 001 | 010 | 100 | Σ | 001 | 010 | 100 | Σ | 001 | 010 | 100 | Σ |
| 40 | 33 | 9 | 8 | 50 | 13 | 6 | 12 | 31 | 9 | 10 | 28 | 47 | 21 | 19 | 20 | 60 |

Tab. 4: Position-dependent and –independent probabilities for two wet 8 hour interval in the uniform splitting (mean of all 24 stations for lower volume class, all values in percent [%]). The combination of '1' (wet) and '0' (dry) illustrates the order of wet and dry 8 hour intervals in a day.

| Two wet intervals, position-independent | Two wet intervals, position dependent | | | | | | | | | | | | | | | |
| --- | --- | --- | --- | --- | --- | --- | --- | --- | --- | --- | --- | --- | --- | --- | --- | --- |
| | starting position | | | | enclosed position | | | | ending position | | | | isolated | | | |
| | 011 | 101 | 110 | Σ | 011 | 101 | 110 | Σ | 011 | 101 | 110 | Σ | 011 | 101 | 110 | Σ |
| **35** | 20 | 5 | 9 | **34** | 14 | 9 | 13 | **36** | 9 | 5 | 22 | **36** | 14 | 3 | 14 | **31** |

Tab. 5: Position-dependent and –independent probabilities for three wet 8 hour interval in the uniform splitting (mean of all 24 stations for lower volume class, all values in percent [%]). The combination of '1' (wet) and '0' (dry) illustrates the order of wet and dry 8 hour intervals in a day.

| Three wet intervals, position-independent | Three wet intervals, position dependent | | | |
|---|---|---|---|---|
| | starting position | enclosed position | ending position | isolated |
| | 111 | 111 | 111 | 111 |
| 25 | 17 | 33 | 16 | 9 |

Tab. 6: Relative and absolute error of rainfall characteristics between disaggregated and observed time series (mean for 24 stations)

| | rE [%] | | | | | | rAE [%] | | | | | |
|---|---|---|---|---|---|---|---|---|---|---|---|---|
| | Wet spell duration | Average intensity | Wet spell amount | Dry spell duration | Fraction of dry intervals | Autocorrelation lag 1 | Wet spell duration | Average intensity | Wet spell amount | Dry spell duration | Fraction of dry intervals | Autocorrelation lag 1 |
| **Method A** | -18 | 40 | 14 | 16 | 2 | -4 | 18 | 40 | 14 | 16 | 2 | 6 |
| **Method B** | -17 | 32 | 9 | 11 | 1 | -3 | 17 | 32 | 9 | 11 | 1 | 6 |
| **Method C** | -16 | -35 | -45 | -47 | -3 | 1 | 16 | 35 | 45 | 47 | 3 | 5 |

Tab. 7 The median of *rE* of extreme rainfall events (over all stations and realisations ) for different return periods and disaggregation methods before and after the resampling

| | | rE [%] | | | | | |
|---|---|---|---|---|---|---|---|
| | | **15 min** | | **1 h** | | **2 h** | |
| | **D** | **Before res.** | **After res.** | **Before res.** | **After res.** | **Before res.** | **After res.** |
| | **Tn** | | | | | | |
| **Method A** | 1 | 17 | 4 | 6 | 19 | 2 | 24 |
| | 2 | 13 | -3 | -2 | 7 | -4 | 12 |
| | 5 | 10 | -9 | -8 | -2 | -7 | 4 |
| | 10 | 8 | -11 | -10 | -6 | -9 | 0 |
| **Method B** | 1 | 13 | 2 | 4 | 17 | 0 | 21 |
| | 2 | 10 | -4 | -3 | 6 | -5 | 11 |
| | 5 | 8 | -9 | -7 | -2 | -8 | 3 |
| | 10 | 7 | -11 | -9 | -6 | -9 | 0 |
| **Method C** | 1 | 10 | 3 | 3 | 15 | -1 | 18 |
| | 2 | 8 | -2 | -4 | 4 | -5 | 8 |
| | 5 | 6 | -5 | -8 | -3 | -9 | 2 |
| | 10 | 5 | -7 | -10 | -7 | -10 | -1 |