# Peer review of "Temporal rainfall disaggregation using a micro-canonical cascade model: Possibilities to improve the autocorrelation"

_Hydrology and Earth System Sciences, 2019_

## Referee Comment (RC1) · Anonymous Referee #1 · 14 Jun 2019

The objective of the paper is to compare different versions of a rainfall disaggregation model that aims to produce high resolution times series (10min) from daily time series of precipitation. The versions is applied on a set of 24 recording stations. The main challenge for the author is to reproduce well the autocorrelation that was observed in the measurements.

This issue is obviously of high interest for the specific configuration of hydrological design in rapid response catchments, especially in urban areas. The manuscript however suffers from a number of limitations. The model is very rough and some basic assumptions of it do likely not hold. This make the model likely poorly relevant. On the other

hand, other disaggregation models have been proposed in past years and the present work should at least include some in their comparison (it just compares variants of the present model but those are not really convincing). The description of the model / results is often rough and requires improvement. I could not understand what is done with some model variants and with the "resampling" process.

I would thus suggest rejection with the recommendation for a resubmission after clarification / improvements. For the Editor, it is not comfortable to have a numbering of lines which is reset to zero at every page. A unique numbering would make the review more convenient.

In the introduction, the authors mention that cascade models underestimate autocorrelation. This is not always true. See the comparative study of Hingray and Ben Haha (2005). They present results obtained for different disaggregation models including the so-called pattern-based microcanonical model. The reproduction of the autocorrelation with this model is almost perfect. This model should likely be included in the present work for comparison. Other models mentioned in the introduction perform also relatively well for AC reproduction. The best ones should be at least included in the present comparative work. The author state in their introduction (p.1 – ln 21/22) that "Since time series with 1280 minutes do not exist as observations, these studies [the studies related to the other models] are rather theoretical than practical from an engineering point of view." It seems to be the reason why the author disregarded the related models. This statement does obviously not hold. All the suggested models can be of high practical interest even from an engineering point of view. You just have to push the disaggregation process at the right temporal level (as the author does it actually in the present disaggregation process > disaggregation to 2.5min + reagregation to 5min).

Relevance of the present model: Variant A : in the first disaggregation step, the daily amount is distributed uniformly on the wet times steps (the wet 8hrs time step can be 1, 2 or 3). This is obviously not realistic at all. The model should relax this strong assumption which obviously cannot be validated from observations (or if it can, this has

to be shown) The amount generated at the 7.5min time step are distributed uniformily on 2.5min time steps and then aggregated back to 5min. The uniform assumption is again really strong. Why don't you do the disaggregation to a finer resolution (3.75 min) and then aggregate back (using the same disaggregation model than the one you used for the previous time step) ?

Variant C : Clarify. I can not understand how it works. The scheme of Figure 2 is very unclear. I do not understand at all.

Avoiding time steps with too small rainfall intensities. Two approaches are considered to tackle this issue. This makes the paper rather complex. The results obtained with both approaches differ not a lot. I would suggest to keep only one of both (The one that mimic the measurement device would be likely relevant).

Resampling. What do you do with the resampling step? Please clarify. What is the archive of observed structure you use ? Please clarify. Give perhaps a graphical scheme for illustration. P10 ln 12. Which structure ? what are the volume classes ? P10 ln 16. "Restriction b) is fulfilled by swapping only relative diurnal cycles as time series elements, which does not affect the daily rainfall amount." I do not understand. Clarify. Why should you swap structure from one day to another ? what about the configuration where the rainfall event lasts more than one day ? do you swap 2-days structures ? if no, why not ?

Definition of an event : P4. Ln 23. Why should you define "events". The separation of events is always rather subjective and all results would depend on the separation rules. Here, you consider that "An event is hereby defined as a wet period enclosed by at least one time step without rainfall before and after the wet period." What is the time step there ? This definition seems to be not really relevant if it is 10mn or 1hour... A number of events present intensities interruption. We cannot consider that a break of one or 2 hours makes different events.

Minor comments : P8 : clarify : is model B non preceding model ? model C : preceding

model ?

P9 ln 2 : what is the "so-called non preceding" ??

p7 ln 20 Clarify : how works the bounded cascade

Equation 2 and equation 4 : what is the sum of probabilities ?

How many parameters have to be estimated for each model ? A table is required.

Can you precise what is an event based and a continuous based evaluation ?

P 11. Ln 13-15. This statement has to be justified

P11. Ln 18-19. "The return period Tn=1.5 years is assumed to be representative for typical return periods for dimensioning purposes in urban hydrology (Tn={1, 2, 5, 10 years},". This can not be possible T15 can not be representative of other T.

p11. Ln 23 : the amount of diurnal cycle ? what is this ?

p12. Ln 23 : for the sake of completeness. I do not see why this is completeness there

p13 : what is partial duration series ??? is it a standard terminology ?

p13 : ln 22 : what is "the single out of all n realisations "?

Table 1 : what are AC values (to be given in the table)

Table 2 : can not understand what is presented there

Table 3 and 4 and 5 : which model ? what should be the sum of probabilities ?

Table 4 and 5 : they have the same caption !!!

Table 6 / 7 : what is the variability between stations ?

Reference : Hingray, B., Ben Haha, M. (2005). Statistical performances of various deterministic and stochastic models for rainfall series disaggregation. Atm. Res. 77:152-175.

---

## Referee Comment (RC2) · Elena Volpi (Referee) · 18 Jun 2019

**1  Summary**

The manuscript deals with micro-canonical Multiplicative Random Cascade (MRC) application for rainfall disaggregation (starting from the daily scale) at sub-hourly scales. Specifically, the Author investigates the effectiveness of some modifications to MRC model proposed by Müller and Haberlandt (2018) in correctly reproducing the autocorrelation function of observed rainfall sub-hourly time series and the occurrence of small rainfall values. This topic is of interest for many hydrological applications; however, the

manuscript presents some weaknesses that should be addressed before the paper can be considered for publication in HESS. My comments are reported in the following; I hope they will be helpful for manuscript improvement.

With appreciation,

Elena Volpi

**2   General comment**

The first concern is related to the motivation of this work. In the Abstract and Introduction Section two main issues that limit the applicability of "classical" MRC models for rainfall disaggregation are mentioned, namely the underestimation of the autocorrelation function of the fine scale process and the presence in the disaggregated series of very small rainfall values, that are not observed in measured data. The first problem does not seem to be of general interest, i.e. it does not characterize all the MRC models, but only the reference model here (that proposed by Müller and Haberlandt, 2018); the second problem is related to the properties of the observed data, which are characterized by a finite resolution so that values smaller than this resolution cannot be recorded. This second problem seems to affect significantly the estimation of the autocorrelation function, as also demonstrated by the results of this work, so that the simulated autocorrelation function does not correspond to the observed one. Hence, the second issue seems to be the fundamental one to improve the effectiveness of the reference model in reproducing the characteristics of the observed data.

Further, only at the end of Introduction Section another fundamental problem characterizing MRC model is mentioned, that is the stationarity of the disaggregated process. The Author cites the paper from Lombardo et al. (2012) where the stationarity issue of MRC model is discussed and an alternative model is proposed that is proved to be stationarity. The Author drew inspiration from the latter model, yet not to guarantee

stationarity but only to improve the reference model performance in terms of autocorrelation function. This is the main issue here; based on my opinion, stationarity should be addressed before improving the accuracy of the simulated autocorrelation function.

Finally, I personally believe that the manuscript is rather difficult to follow because most of the mathematic behind the models is not explained (see the specific and technical comments that follow). As an example, it is not clear how many parameters rule the model behavior (in its different modified versions) and how these parameters can be estimated starting from the observed data, even if it seems that in this case the Author does not assume an universal generator (hence a simple-/multi-scaling behavior), am I correct?

**3   Specific and technical comments**

Lines 14-17, page 2. The difference between canonical and micro-canonical is that between downscaling and disaggregation, as pointed out in Koutsoyiannis and Langousis (2011).

- Koutsoyiannis, D. and Langousis, A. (2011). Precipitation. In Treatise on water science, Edited by: Wilderer, P. and Uhlenbrook, S. Vol. 2, 27–78. Oxford: Academic Press.

Lines 14-17, page 2. This seems to be a minor problem, which is in general solved by disaggregating at a finer time scale and then aggregating at the desired one. It might have some implications in parameter estimation, depending on the structure of the generator. Is this the case?

Lines 17-19, page 3. Among the problems that the Author cites that justify the modifications of the MRC proposed in the manuscript, there is the non-stationarity issue.

However, this is not mentioned in the abstract, but only at the end of the Introduction Section, and nowhere else in the manuscript; in other words process stationarity is not considered a problem here (see also the general comment).

Lines 19-20, page 3. The description of the work by Lombardo et al. is not clear and, based on my opinion, what emerges does not correspond to the work done in the cited papers. The Author should improve his synthetic explanation; further, note that the method proposed by Lombardo et al. is not based on a MRC, but on an "additive" cascade.

Lines 26-28, page 3. I'm not sure I fully understood the issue of small values that are generated by the random cascade. Are the small values too small with respect to those characterizing observed rainfall time series at the target temporal resolution? If the reference truth is the observed high-resolution time series, is the Author arguing that the reference truth is not correct? This issue, which constitutes one of the most important motivations of this study, should be better explained to the potential readers (see also the general comment).

Lines 13-14, page 5 and 1-2, page 6. The sentence is not clear.

Line 4, page 6. "direction"?

Line 11, page 6. What does it mean that the aim is achieving a "minimum rainfall intensity"?

Line 12, page 6. Resampling as a subsequent step after disaggregation could be define as a post-processing technique/strategy.

Line 24, page 6. "no" instead of "on"?

Eq. (2). Are the possible outcomes for the three "events" at the disaggregation level 2 all mutually exclusive? In such a case, should the sum of the corresponding probabilities be equal to 1? If this is not the case, the disaggregation scheme at level 2 should be better explained.

Line 4, page 7. What does it mean that the (probability?) parameters depend on the volume? How? Even if this is explained in a previous paper, it should be briefly recalled here for the sake of clarity.

Lines 15-20, page 7 and 1-2, page 8. How many parameters characterize this modified version of the model? Which is the rule of dependence for starting, enclosed, etc. elements in the disaggregated series? The explanation is incomplete, or at least too vague.

Section 3.2.1. Method B adds additional probability parameters, letting them vary with the "position". It is unclear how these multiplicative parameters are determined to reproduce the statistical characteristics of the process across scales based on the structure of the cascade (i.e. b=3 for level one and b=2 for all the remaining levels up to the desired temporal resolution). See also previous comment.

Figure 2 could be improved to help for reader understanding the difference between method B and C (but also starting, ending etc. elements of the cascade). Further, it should be explicitly mention if blue and white denote wet and dry states.

Line 11, page 9. "isolated" instead of the second "ending"?

There is a substantial difference between the approach proposed here and that used in Lombardo et al. (2012, 2017). Here the Author introduces a "conditional" probability that determines the probability of a wet or dry state, while in Lombardo et al. the information of previously disaggregated elements at the same time-resolution is used to feed a linear interpolation model estimating the disaggregated value of the subsequent element (see eq. (7) in Lombardo et al., 2017).

Line 7 page 10. The underestimation of the autocorrelation function characterizing the starting model is not a characteristic of all multiplicative random cascade models, as stated by the Authors in the literature review. Is there a motivation for this? Is it possible to generalize the problem to a specific MRC model (i.e. generator) structure?
Line 17, page 14. If the reference model works well in terms of autocorrelation, that is the main issue here, why two different modifications are proposed?

I do understand that the dataset is not so rich, but I expected to find two separated datasets, one for "calibration" and one for "validation" of the models. The Author should justify this choice.

––––––––––––––––––––––––––––

---

## Author Comment (AC2) · 21 Jun 2019

Please find my reply in the supplement.

Please also note the supplement to this comment:
https://www.hydrol-earth-syst-sci-discuss.net/hess-2019-216/hess-2019-216-AC2-supplement.pdf

---

## Referee Comment (RC3) · Anonymous Referee #3 · 22 Jun 2019

The paper deals with the improvement of a given disaggregation model using micro-canonical cascade. In general the topic is relevant for the community. The paper is interesting, but cannot be published in its current state, and requires major modifications.

General comments: - The paper is quite hard to read with many models being compared. Explanations for the slight variations between the various models are sometimes hard to follow. There is a lack of mathematical details in the presentations of the various models. - Only comparison between variations of a given model are provided. Comparisons with other type of cascade models should at least be discussed. - There

are numerous parameters to be estimated per model (not even very clear which number according to the model choice). It is not clear whether a calibration period and a validation period were used.

Detailed comments:

1) Introduction - p.2 l.21 : "since time series with 1280 minutes do not exist as observation do not exist". I do not understand this statement and this does not seem a real issue. Anyway, if needed, you can disaggregate at a higher resolution and up-scale to the desired one.

2) Rainfall data - p. 3 l.28-29 : "from a practical. . . have an impact on the autocorrelation function". Why not trying the compute the autocorrelation using higher moments to limit the influence of smaller values ? - p.4 l.10 : "how can a minimum rainfall intensity be ensured during the disaggregation process?". It is not very clear to me the need for this, since as pointed out by the author and references cited, it might very well be simply due to the rain gauge measurement limitations. It might be worth testing a time series obtained with a disdrometer which enables better representation of small values of rainfall.

3) Methods - p.5 l.8 : "actual" - - > "Actually" - Eq. 1 : it should be added how lambda is related to t1 and t2 (I guess lambda = t2-t1) - p. 6 l.24 : "on" - - > "no" ? - Eq. 2 : shouldn't it be P(0/0/1)/3 in the first line since there are three possibilities (100,010,001) for the same P(0/0/1) ? This remark is also valid for all the other probabilities except P(1/3,1/3,1/3) - p.7 l.11 : "an empirical function", please be more specific (see also general comment on the lack of mathematical details). - p.7 l.15-18 : a summary table or scheme would be helpful. - It remains weird to have different branching number and probabilities weights for the first cascade steps which seems to be in contradiction with the underlying scaling properties. - Section 3.1.3 : I found the paragraph quite hard to read. may be a more precise scheme could be helpful. It should be mentioned that it adds a lot of parameters. In general, a summary table with the number of parameters

according to the model would be helpful. - Section 3.2 : why presenting two different models (especially given that they provide rather similar results) ? It adds complexity to a paper with already a lot of comparison. I would keep only the MMD which is the more realist I believe. - Section 3.3 : The process with Ir and more generally the swapping seems rather ad hoc. It seems that the underlying physical meaning of cascade process is lost. I think that this issue should at least be discussed. - p. 13 l. 18 : "30 realisations". Why such a small number, it seems that much more could have been performed.

4) Results - Table 3 and 4 are really hard to follow. I think a scheme representing the various cases could be really helpful. - Why the average rainfall intensity changes is such a micro-canonical cascade ? - p. 19 l. 5-10 : may be a graph showing the sensitivity of the results to Ir would be be needed.

5) Discussion - p. 21 l. 14-15 : "identified similarities. . . used for simplification", please clarify and be more explicit.

---

## Author Response (AR1)

**General Reply**

I thank all three reviewers for their constructive comments on the manuscript, which helped to improve it significantly. Please find detailed replies to all comments on the following pages. These replies were uploaded very fast to encourage a

- 5 discussion in the open review process. However, after the open review phase is closed now an additional focus was given to increase the readability of the manuscript. The following steps were carried out:
  - (i) The readability was increased by adding a flowchart (Fig. 2) illustrating all applied methods, a notation where to find their description in the manuscript and the resulting data sets.
  - (ii) Additional information are provided at the beginning of sections including method descriptions for an easier navigation.
  - (iii) Although I have rejected it before, I followed now the suggestion of two reviewers to leave out the comparison between MRA and MMD. Only the MMD approach is still included in the manuscript.

With these changes the readability of the manuscript should have been increased and the navigation should be (hopefully) easier.

**15**

20

10

**Review by anonymous Referee #1**

The objective of the paper is to compare different versions of a rainfall disaggregation model that aims to produce high resolution times series (10min) from daily time series of precipitation. The versions is applied on a set of 24 recording stations. The main challenge for the author is to reproduce well the autocorrelation that was observed in the measurements.

This issue is obviously of high interest for the specific configuration of hydrological design in rapid response catchments, especially in urban areas. The manuscript however suffers from a number of limitations. The model is very rough and some basic assumptions of it do likely not hold. This make the model likely poorly relevant. On the other hand, other disaggregation models have been proposed in past years and the present work should at least include some in their

- 25 disaggregation models have been proposed in past years and the present work should at least include some in their comparison (it just compares variants of the present model but those are not really convincing). The description of the model / results is often rough and requires improvement. I could not understand what is done with some model variants and with the "resampling" process.
- 30 I would thus suggest rejection with the recommendation for a resubmission after clarification / improvements. For the Editor, it is not comfortable to have a numbering of lines which is reset to zero at every page. A unique numbering would make the review more convenient.
- I thank the reviewer for the effort and the time spend on this manuscript. The reviewer points out the importance of the study for urban hydrological applications. His/her major concerns are an insufficient model description and missing comparisons of the results with other references. For both concerns detailed examples are provided by the reviewer. A point-by-point reply can be found below. All page and line numbers refer to the original submission.

**Specific comments:**

40 1. The objective of the paper is to compare different versions of a rainfall disaggregation model that aims to produce high resolution times series (10min) from daily time series of precipitation. The versions is applied on a set of 24 recording stations. This comment is copied from the general part of the review for clarification. The disaggregated time series have a final temporal resolution of 5 min, not 10 min. Although the observed time series have a temporal resolution of 5 min (see e.g. P4L20, P8L3-5, P13L15-16,...), it seems the information was missing at other parts of the manuscript. Hence, the information was added in the abstract (P1L9-10):

5 "In this paper two cascade model modifications are analysed regarding their ability to improve the autocorrelation in disaggregated time series with 5 minute resolution."

and in the introduction of the method section (P5L5):

"The overall aim of this investigation is the improvement of the autocorrelation  $r_{tl,t2}$  of the generated time series with a temporal resolution of 5 minutes."

10

2. In the introduction, the authors mention that cascade models underestimate autocorrelation. This is not always true. See the comparative study of Hingray and Ben Haha (2005). They present results obtained for different disaggregation models including the so-called pattern-based microcanonical model. The reproduction of the autocorrelation with this model is almost perfect. This model should likely be included in the present work for comparison.

- 15 The reference of Hingray and Ben Haha (2005) was not included in the review of autocorrelation results so far. However, it was not the intention to "hide" results which do not fit to the motivation for the current study. Indeed, it is pointed out that under- and overestimations can occur by the application of a micro-canonical cascade model (P2L29-P3L4) due to several reasons (P3L5-10). Hingray and Ben Haha use for their investigation the time series of only one rain gauge and over a short range of disaggregation levels. Despite the low representation using only one rain gauge, by the few disaggregation levels it
- 20 can be expected to achieve good results in terms of autocorrelation. The disaggregation starts with observed hourly values (so on this level the autocorrelation is "perfect", because it results from observations) and ends after four disaggregation steps (start: 1 h, 30 min, 15 min, 7.5 min...even less, since afterwards the time series is aggregated back to 10 min time steps). To keep the "perfect" autocorrelation over four disaggregation steps is much easier than over 7 disaggregation steps (start: 1d, 8 h, 4 h, 2 h, 1 h, 30 min, 15 min, 7.5 min [5 min]) as done in the current study. It can be questioned if the
- 25 autocorrelation would be as good represented by a disaggregation of daily values down to 5 min values. Also, the results are shown in Hingray and Ben Haha only for lag-1 autocorrelation. As shown in the current study, although a good fit can be achieved for lag-1, strong underestimations can occur for other lags (see e.g. Fig. 6, Method C). Hence, I have implemented the reference into the literature review, but do not attempt to apply the disaggregation model additionally for the sake of comparison (especially in accordance with comment 8 of the reviewer, that the manuscript is already complex).

**30**

The following sentences were implemented after P3L4:

"A good representation of the lag-1 autocorrelation was achieved by Hingray and Ben Haha (2005) with two microcanonical cascade models. However, since only four disaggregation steps were applied (from hourly to 7.5 min time steps) it remains unclear, if the good representation would have been achieved for more disaggregation steps."

35

3. Other models mentioned in the introduction perform also relatively well for AC reproduction. The best ones should be at least included in the present comparative work.

I double-checked the references cited in the introduction regarding autocorrelation results. As far as I can see, the references not discussed do not include autocorrelation results (e.g. Molnar and Burlando, 2005, Licznar et al. 2011, 2015). The autocorrelation results of Paschalis et al. (2014) are not discussed, since they result from combination of different rainfall generators, thus the error cannot be assigned exactly to the micro-canonical cascade model or the others. However, for the majority of rain gauges and rainfall generator combinations, the lag-1 autocorrelation is also underestimated. Since the reviewer mentioned no reference explicitly, it remains unclear for me which results should be discussed additionally.

A comparison with other rainfall generators is way beyond the scope of the current study, which aims at improving the general model structure. The basic idea behind the preceding cascade model and the resampling approach can be transferred to other micro-canonical cascade models, so this study is considered more as a methodological development than a bestchoice model comparison. 4. The author state in their introduction (p.1 - ln 21/22) that "Since time series with 1280 minutes do not exist as observations, these studies [the studies related to the other models] are rather theoretical than practical from an engineering point of view." It seems to be the reason why the author disregarded the related models. This statement does obviously not hold. All the suggested models can be of high practical interest even from an engineering point of view. You just have to

- 5 push the disaggregation process at the right temporal level (as the author does it actually in the present disaggregation process > disaggregation to 2.5min + reagregation to 5min). The reviewer is right, a disaggregation could have started also on daily values applying the same disaggregation model until the disaggregated time series have a temporal resolution below the desired resolution and then a kind of transformation can
- be applied. However, this was not done in the mentioned studies. As shown in Müller and Haberlandt (2015) the kind of
   transformation has a significant influence on the disaggregation results, e.g. over- or underestimation of the rainfall intensities and especially on extreme rainfall values. Hence, a fair comparison is not possible. However, it was not the intention to disregard these disaggregation models. Hence I added the following subsequent sentence:

"Of course, by the application of an additional transformation process a desired temporal resolution can be achieved, whereby the transformation process affects the characteristics of the disaggregated time series."

15

5. Relevance of the present model: Variant A : in the first disaggregation step, the daily amount is distributed uniformly on the wet times steps (the wet 8hrs time step can be 1, 2 or 3). This is obviously not realistic at all. The model should relax this strong assumption which obviously cannot be validated from observations (or if it can, this has to be shown)

The reviewer is right, the uniform distribution over the as wet identified 8 h-time steps is a strong assumption. Although the number of wet 8 h-time steps is realistic, the resulting rainfall amounts are maybe less realistic. There are two reasons for this assumption.

First, the number of wet 8 h-time steps tells us something about the genesis of the rainfall event. If it is only one wet time step, it is likely a convective event and hence the whole daily rainfall amount is put into this single 8 h-time step. If the rainfall event lasts longer than 8 h, it is likely a stratiform event, so a long-lasting and less intense event (with often more or

25 less uniform rainfall occurrence). The uniform distribution can be found only on the 8 h-level, on finer temporal resolutions intensities vary due to the b=2-splitting so the resulting final time series does not show any uniform rainfall intensity distribution anymore.

Second, the assumption with the uniform distribution demands only a few parameters. For method A, two parameters are required (P(0/0/1) and P(0/0.5/0.5)). The only other tested approach with a b=3-splitting in the first disaggregation step was introduced by Lignick et al. (2012), who use for example 8 distribution functions for a grifting with 2 wet 8 h integrals.

- 30 introduced by Lisniak et al. (2013), who use for example 8 distribution functions for a splitting with 3 wet 8 h intervals. To keep the cascade model parameter parsimonious, the uniform distribution has been chosen. Also, in previous publications (Müller and Haberlandt, 2015, 2018, Müller-Thomy et al., 2018) with this assumption good representation of rainfall characteristics have been achieved.
- 6. The amount generated at the 7.5min time step are distributed uniformily on 2.5min time steps and then aggregated back to 5min. The uniform assumption is again really strong. Why don't you do the disaggregation to a finer resolution (3.75 min) and then aggregate back (using the same disaggregation model than the one you used for the previous time step) ? The idea of the reviewer is to apply a different transformation to achieve a final temporal resolution of 5 min. In a previous publication Müller and Haberlandt (2018) tested transformations at 3 different temporal resolutions, starting with 15 min,
- 40 7.5 min and 3.75 min, for the same data set. The transformation starting with 7.5 min led to the overall best results. An alternative would be to continue with the disaggregation to a very fine temporal resolution (few seconds) and then aggregating the time steps to 5 min, so that there is no need for any transformation. However, since a bounded cascade model is applied, the parameters for this so-called 'fine-graining' process have to be estimated from observations with the same temporal resolution, which are not available for the current study.
- 45

7. Variant C : Clarify. I can not understand how it works. The scheme of Figure 2 is very unclear. I do not understand at all. For a better understanding, the disaggregation level and temporal resolutions have been added to Fig. 2 and colours have been removed. Unfortunately, without a certain comment what is unclear for the reviewer (the position definition, the time

step indices, the increase in number of position classes or what exactly in the figure remains unclear) the manuscript can hardly be improved further in this point.

8. Avoiding time steps with too small rainfall intensities. Two approaches are considered to tackle this issue. This makes the
paper rather complex. The results obtained with both approaches differ not a lot. I would suggest to keep only one of both (The one that mimic the measurement device would be likely relevant).

The reviewer suggests to leave the MRA approach out to reduce the complexity of the manuscript. However, both approaches represent possible solutions and none of them has been tested before to the authors knowledge. To reduce the complexity, only the MMD approach has been applied for the resampling investigations. I prefer to keep both approaches in

10 the current study, especially since reviewer 2 identifies this issue as the more fundamental one. The comparison of the two approaches is also useful for other researchers, because they know that the outcome is very similar and don't have to carry out a study on their own.

**8. Resampling. What do you do with the resampling step? Please clarify.**

15 The introduction of subsection 3.3 has been extended to clarify, what is done in the resampling algorithm (which is explained more in detail afterwards). The following sentence has been added after P10L7:

"In a resampling process, two elements (here: relative diurnal cycles of the disaggregated time series) are swapped to improve an objective function (here: minimizing the deviation of the autocorrelation function of the disaggregated time series from the observed time series)."

20

**9. What is the archive of observed structure you use ? Please clarify. Give perhaps a graphical scheme for illustration.**

For the resampling, there is no archive of observed structures. The structure of the disaggregated time series results solely from the disaggregation process. After the disaggregation, two relative diurnal cycles of the disaggregated time series are swapped with the aim to improve the autocorrelation function (see point 1 of the resampling scheme, P12L2-3).

25

**10. P10 ln 12. Which structure ? what are the volume classes ?**

The reviewer refers to the sentence: "The structure of position and volume classes in the disaggregated time series generated by the cascade model should be conserved." Indeed, the sequence of position (e.g. starting, enclosed, ending, isolated, see Fig. 3) and volume classes combination (lower and upper volume class) of a day defines the structure of the disaggregated time series.

30 time series.

Both type of classes are introduced on P7L15-18.

11. P10 ln 16. "Restriction b) is fulfilled by swapping only relative diurnal cycles as time series elements, which does not affect the daily rainfall amount." I do not understand. Clarify.

35 One advantage of the micro-canonical cascade model is that the daily rainfall amounts are conserved exactly. If for the resampling absolute diurnal cycles are swapped, daily totals will differ from the original coarse time series, which has been disaggregated. To avoid this issue, no absolute values, only relative diurnal cycles are swapped. To achieve a "relative" diurnal cycle, for each 5 min time step its fraction of the total daily rainfall amount is determined. This sequence of fractions are swapped and subsequently multiplied with the rainfall amount of the other day chosen for the swap. Hence, the daily rainfall amount remains constant for each day.

**12. Why should you swap structure from one day to another ?**

The aim of the resampling is to improve the autocorrelation function (P10L7). Structures are not swapped, only relative diurnal cycles. The structure of the time series is kept.

**45**

13. what about the configuration where the rainfall event lasts more than one day ? do you swap 2-days structures ? if no, why not ?

If rainfall lasts for more than one day, this is taken into account by the position classes. For example, in three consecutive days there is {no rain; rain; rain}, the rainfall of the second day (starting position) can be most likely found at the end of the

day after the disaggregation process due to the position dependency of the parameters. This is the content of Table 3, 4 and 5 and explained at P14L3-11. To answer the question, the relative diurnal cycle of only one day is swapped, but the long-lasting event should not be destroyed by the swap.

- 5 14. Definition of an event : P4. Ln 23. Why should you define "events". The separation of events is always rather subjective and all results would depend on the separation rules. Here, you consider that "An event is hereby defined as a wet period enclosed by at least one time step without rainfall before and after the wet period." What is the time step there ? This definition seems to be not really relevant if it is 10mn or 1hour. A number of events present intensities interruption. We cannot consider that a break of one or 2 hours makes different events.
- 10 The reviewer questions the event definition in the manuscript. Indeed, the event definition is always subjective and from the authors opinion there is no right or wrong for this case. The reviewer argues that 5 min dryness are not long enough for a separation, but also 2 hours are not long enough in the reviewers opinion. A plausible event definition would differ between events, since it depends on the kind of event (for a convective event with a duration of few minutes or hours the separation time would be much shorter than for stratiform events, which last over several days). For this kind of validation much more
- 15 information than the rainfall time series would be required, e.g. circulation patterns or air pressure time series. But this would be way beyond the scope of the current study. Since there is no "true" separation time" as the reviewer points out herself/hisself, I prefer to keep the 5 min time step for separation as an objective criteria. For clarification, the sentence has been updated to:
- "The definition for a single event is according to Dunkerley (2008); having a minimum of one dry **5 min** time step before and after the rainfall occurrence."

**Minor comments:**

P8 : clarify : is model B non preceding model ? model C : preceding model ?

25 Method C is the only cascade model modification which is considered as "preceding" cascade model. For clarification purposes, the following sentence has be added after P8L18:
"Since only method C is referred to as preceding associate model, method A and P can be considered as non-preceding.

"Since only method C is referred to as preceding cascade model, method A and B can be considered as non-preceding cascade models."

30 P9 ln 2 : what is the "so-called non preceding" ?? Please see the reply to your former comment.

**p7 ln 20 Clarify : how works the bounded cascade**

- The differentiation into "bounded" and "unbounded" relates to the range of temporal scales used for the estimation of the cascade model parameters. In an unbounded cascade model it is assumed, that the parameters are similar over the applied range of temporal scales due to a mon-fractal scaling behaviour and hence the same parameter set can be applied for all disaggregation steps (e.g. from daily values to hourly time steps). In a bounded cascade model, the parameter differ between the disaggregation levels (e.g. the parameter set applied for the disaggregation from 8 h to 4 h is different than from 10 min to 5 min) to take into account the multi-fractal scaling behaviour, so for each disaggregation step a particular parameter set is applied (which is 'bound' to the certain disaggregation step). The explanation on P7L20:
- "For each step of the disaggregation process a particular parameter set is applied, which is 'bound' to the specific transition of temporal resolution."

has been extended to:

to clarify this issue.

"For each step of the disaggregation process a particular parameter set is applied, which is 'bound' to the specific transition 45 of temporal resolution. The need for particular parameter sets for each disaggregation step arise from the wide temporal range (from daily to 5 min time steps) and hence the underlying processes covered causing multi-fractal scaling behaviour."

Equation 2 and equation 4 : what is the sum of probabilities ?

The sum for each line of the equations is '1' as mentioned at P6L1 for Eq. 2:

"The sum of the weights is equal to 1 in each split, so the rainfall amount is conserved exactly throughout the disaggregation process."

For clarification purposes, this information has been repeated for Eq. 4: 5

"Again, all probabilities (P(0/1), P(1/0) and P(x/(1-x))) sum up to 1. "

**How many parameters have to be estimated for each model ? A table is required.**

I agree with the reviewer, a new table (Table 2) has been implemented, which lists the parameter amount for each branching type and cascade model variant exactly.

10

Can you precise what is an event based and a continuous based evaluation ?

For an event-based evaluation first events are separated and then the characteristics of these events are determined. For a continuous-based evaluation the whole time series is considered, event-independent. The two sentences written bold has been added for explanation (at P13L1-5):

**15 "For an event-based evaluation, first the rainfall events are identified and then the characteristics of these events are**

determined. Event-based rainfall characteristics include wet and dry spell duration as well as wet spell amount. An event is

hereby defined as a wet period enclosed by at least one time step without rainfall before and after the wet period.

For a continuous-based evaluation, the whole time series is considered, without differentiation into single events. As

continuous time series characteristics, the average intensity, the fraction of dry intervals and the autocorrelation are

20 analysed."

**P 11. Ln 13-15. This statement has to be justified**

The statement is justified by the citation of Müller and Haberlandt (2018). Additional explanations as e.g. deviations of flooding volume or combined sewer overflow volume would require an additional brief description of the investigated 25 sewage system to plausibilize these values (independent if absolute or relative deviations would be shown). This would move the scope of the paragraph to a wrong direction. Hence, no additional information has been added for justification.

P11. Ln 18-19. "The return period Tn=1.5 years is assumed to be representative for typical return periods for dimensioning purposes in urban hydrology ( $Tn = \{1, 2, 5, 10 \text{ years}\}$ ,". This can not be possible T15 can not be representative of other T.

- The reviewer points out a sentence, which is indeed misleading. Of course, the absolute values of Tn=1.5 years are different 30 from other return periods. The intention was to express, that an under- or overestimation of observed values for Tn=1.5 yrs is similar for these return periods, since they are close to each other from a statistical point of view. Of course, for a return period of Tn=100 years additional analysis would have to be carried out. The sentence was rephrased to:
- "It is assumed that the results regarding under- or overestimation for the return period  $T_n=1.5$  years are representative for typical return periods for dimensioning purposes in urban hydrology ( $T_n = \{1, 2, 5, 10 \text{ years}\}$ , DWA-A 531 (2012))." 35

**p11. Ln 23 : the amount of diurnal cycle ? what is this ?**

The word "amount" was wrong here. It was changed to "number".

**40 p12. Ln 23 : for the sake of completeness. I do not see why this is completeness there**

Without the parameter values it would be only a method description, not exactly repeatable. With the absolute values a repetition is possible, because all information are provided. However, the sentence has been changed to:

"For the sake of completeness the The following setup was chosen for the resampling:  $T_{a,start}=1*10^{-4}$ , dt=0.99, K=500,  $M=200 \text{ and } thr_{O,auto}=1*10^{-9}$ ."

**p13 : what is partial duration series ??? is it a standard terminology ?**

Yes, at least in Germany it is a standard terminology. However, the subsequent sentence includes the information that it is similar to the peak-over-threshold approach:

"Partial duration series are similar to the peak-over-threshold approach, whereby the threshold is defined in order to select 3 extreme rainfall events on average per year."

**p13 : ln 22 : what is "the single out of all n realisations "?**

The sentence has been rephrased to:

"with *i* as control variable of all realisations *n*:"

10

Table 1 : what are AC values (to be given in the table)The autocorrelation values for lag-1 have been added to Table 1.

**Table 2 : can not understand what is presented there**

15 The caption of the table was changed to "Nomenclature of dataset abbreviations used in this study". The intention of the table is to provide an overview, which combinations of data sets were investigated in the study.

**Table 3 and 4 and 5 : which model ?**

The first column of each table ("position-independent") belongs to method A, while all other columns belong to method B ("position-dependent).

The sentence:

"The resulting probabilities are shown in Table 3, 4 and 5." (P14L3) has been extended to:

"The resulting probabilities are shown in Table 3, 4 and 5 (columns with position-dependent entries) in comparison to the

25 position-independent probabilities estimated for method A (first column in each table).

**what should be the sum of probabilities ?**

The sum of probabilities has to be 1 as defined on P7L1-2: "The sum of the weights is equal to 1 in each split, so the rainfall amount is conserved exactly throughout the disaggregation process." For example, the sum of the probabilities for one wet interval, position-independent (Tab. 3, first column, first entry) P=40 %, for two wet intervals, position-independent (Tab. 4) P=35 % and for three wet intervals, position-independent (Table 5) P=25 % is ∑=100 %. Small deviations from 100 % can

occur, since the mean for 24 stations is shown and hence rounding errors can occur.

**Table 4 and 5 : they have the same caption !!!**

35 The reviewer is right, for Table 5 it should have been "...three wet 8 hour interval..." instead of "...two wet 8 hour interval...". This has been corrected.

**Table 6 / 7 : what is the variability between stations ?**

In Table 6 and 7 the relative errors for rainfall characteristics and extreme values are presented. These error variability differs between the investigated methods. However, I think an information about the standard deviation of the relative error or something similar is not useful for the reader and hence it was not added to the tables.

- characterize all the MRC models, but only the reference model here (that proposed by Müller and Haberlandt, 2018); the second problem is related to the properties of the observed data, which are characterized by a finite resolution so that values smaller than this resolution cannot be recorded. This second problem seems to affect significantly the estimation of the autocorrelation function, as also demonstrated by the results of this work, so that the simulated autocorrelation function does not correspond to the observed one. Hence, the second issue seems to be the fundamental one to improve the effectiveness of the reference model in reproducing the characteristics of the observed data.
- From the literature review it is obvious that the reproduction of the autocorrelation function is a general problem of the micro-canonical cascade models, whereby under- and overestimations can occur. The introduced position definitions and the resampling approach were proven to solve the underestimation and it is assumed, that an overestimation would be reduced by both approaches as well. Both approaches are transferable to other cascade models and hence represent general
- 35 improvements/solutions, which are not restricted to the micro-canonical cascade model applied in this study. Of course, I agree with the reviewer that the generation of too small rainfall intensities is an important issue, especially since all rainfall characteristic comparisons are biased if the time steps with hydrologic irrelevant rainfall intensities are taken into account. Solving this second issue provides the basement for solving the first issue.
- 40 2. Further, only at the end of Introduction Section another fundamental problem characterizing MRC model is mentioned, that is the stationarity of the disaggregated process. The Author cites the paper from Lombardo et al. (2012) where the stationarity issue of MRC model is discussed and an alternative model is proposed that is proved to be stationarity. The Author drew inspiration from the latter model, yet not to guarantee stationarity but only to improve the reference model performance in terms of autocorrelation function. This is the main issue here; based on my opinion, stationarity should be addressed before improving the accuracy of the simulated autocorrelation function.
- Indeed, the problem of stationarity is not solved in this manuscript, but this was also not the intention. The inspiration from Lombardo et al. (2012) for the current study refers "only" to the determination of the most important time steps to generate highly correlated time series. However, the analysed position definitions as well as the resampling approach can be transferred to other cascade models, so also later for stationary models.

3. Finally, I personally believe that the manuscript is rather difficult to follow because most of the mathematic behind the models is not explained (see the specific and technical comments that follow). As an example, it is not clear how many parameters rule the model behavior (in its different modified versions) and how these parameters can be estimated starting

5 from the observed data, even if it seems that in this case the Author does not assume an universal generator (hence a simple-/multi-scaling behavior), am I correct?
The reviewer points out missing information about the method and its implication. A new subsection was added (Section)

The reviewer points out missing information about the method and its implication. A new subsection was added (Section 3.1.4, including a model parameter comparison) as well as a paragraph for the parameter estimation. All detailed comments from the reviewer have been addressed and can be found below. The rainfall generator is described by Eq. 2 (Eq. 4 for method B and C) for b=3 and Eq. 3. for b=2.

**3** Specific and technical comments**

10

 Lines 14-17, page 2. The difference between canonical and micro-canonical is that between downscaling and disaggregation, as pointed out in Koutsoyiannis and Langousis (2011). (Koutsoyiannis, D. and Langousis, A. (2011). Precipitation. In Treatise on water science, Edited by: Wilderer, P. and Uhlenbrook, S. Vol. 2, 27–78. Oxford: Academic Press)

The additional information was implemented in the manuscript:

- "Based on Koutsoyiannis and Langousis (2011), the canonical version of the cascade model (conservation of rainfall amount
   on average during the disaggregation, e.g. Molnar and Burlando, 2005, Paschalis et al., 2012) represents a downscaling technique, while the micro-canonical version (exact rainfall amount conservation for each time step, e.g. Olsson, 1998, Güntner et al., 2001, Licznar et al., 2011, 2015) represents a disaggregation technique."
- 5. Lines 14-17, page 2. This seems to be a minor problem, which is in general solved by disaggregating at a finer time scale and then aggregating at the desired one. It might have some implications in parameter estimation, depending on the structure of the generator. Is this the case?

I assume the comment refers to P2L17-22. The implication of putting a1440 min rainfall amount (1 d) into 1280 min and start the disaggregation is that you will end up with a day which is 160 min (~2.5 h) too short. So either you assume that the missing time steps have a rainfall amount of 0 mm (the question is then: Where to put them?) or you apply any transformation to the daily time series before the disaggregation. Both approaches will affect the resulting rainfall time series

- 30 transformation to the daily time series before the disaggregation. Both approaches will affect the resulting rainfall time series characteristics. The following sentence was added: "Of course, by the application of an additional transformation process a desired temporal resolution can be achieved, whereby the transformation process affects the characteristics of the disaggregated time series."
- 6. Lines 17-19, page 3. Among the problems that the Author cites that justify the modifications of the MRC proposed in the manuscript, there is the non-stationarity issue. However, this is not mentioned in the abstract, but only at the end of the Introduction Section, and nowhere else in the manuscript; in other words process stationarity is not considered a problem here (see also the general comment).
- Of course, non-stationarity is a problem, but it is not the motivation for the analysis carried out in this study. Indeed,mentioning the problem in the introduction was misleading and this sentence has been removed (see also the reply to your comment 20).

7. Lines 19-20, page 3. The description of the work by Lombardo et al. is not clear and, based on my opinion, what emerges does not correspond to the work done in the cited papers. The Author should improve his synthetic explanation; further, note that the method proposed by Lombardo et al. is not based on a MRC, but on an "additive" cascade.

45 that the method proposed by Lombardo et al. is not based on a MRC, but on an "additive" cascade. The reviewer is right, the disaggregation models differ. The related parts of the text have been rephrased, so that only the information of Lombardo et al. is briefly cited, which are the most worth time steps to consider for the generation of highly correlated time series under the burden of computational efforts. A new paragraph has been added to the section "5.3 Study limitation" for clarification: "Fourth, although method C is based on a finding in Lombardo et al. (2012, 2017), the disaggregation method differs from the additive cascade model in Lombardo et al. (2012, 2017). Hence, the by Lombardo et al. identified problem of non-stationarity of the disaggregation is not solved by the introduced cascade model variants and remains an open challenge for further studies."

5 Please see also the reply to your comments 6 and 20.

8. Lines 26-28, page 3. I'm not sure I fully understood the issue of small values that are generated by the random cascade. Are the small values too small with respect to those characterizing observed rainfall time series at the target temporal resolution? If the reference truth is the observed high-resolution time series, is the Author arguing that the reference truth is

- not correct? This issue, which constitutes one of the most important motivations of this study, should be better explained to the potential readers (see also the general comment).
   I thank the reviewer for pointing out this vague lines. This part of the introduction has been rephrased to:
   "Koutsoyiannis et al. (2003) argue that it is unclear, if the values generated by the cascade model are too small in comparison to the observed minimum rainfall intensities or if the resolution of the measurement device is not fine enough to
- 15 observe the very small rainfall intensities generated by the cascade model. From a practical point of view, these low-intensity time steps are not important, but they have an impact on the autocorrelation function. To enable comparisons between the autocorrelation of observed and disaggregated rainfall time series two methods are analysed in this study which ensure a minimum rainfall intensity in the disaggregated time series."
- 9. Lines 13-14, page 5 and 1-2, page 6. The sentence is not clear.The reviewer is right, the sentences were more misleading than helpful. Both sentences were removed from the manuscript.

10. Line 4, page 6. "direction"?

The sentence has been reduced to:

25 "The autocorrelation function is based on two elements: the covariance  $s_{tl,t2}$  of the original and the shifted time series ( $t_l$  and  $t_2$ ), that describes the direction of the relation of both time series, and the standard deviations of both time series,  $s_{tl}$  and  $s_{t2}$ , for the standardization of the covariance."

11. Line 11, page 6. What does it mean that the aim is achieving a "minimum rainfall intensity"?

30 The text has been rephrased to:

"...to achieve the same minimum rainfall intensity as in the observed time series, to enable comparisons between the autocorrelation in observed and disaggregated rainfall time series."

12. Line 12, page 6. Resampling as a subsequent step after disaggregation could be define as a post-processing technique/strategy.

The term "subsequent step" was replaced by "post-processing strategy" throughout the manuscript.

3. Line 24, page 6. "no" instead of "on"?

I thank the reviewer for this error spot, it was corrected.

**40**

14. Eq. (2). Are the possible outcomes for the three "events" at the disaggregation level 2 all mutually exclusive? In such a case, should the sum of the corresponding probabilities be equal to 1? If this is not the case, the disaggregation scheme at level 2 should be better explained.

Indeed, in Eq. (2) the occurrence of rainfall amounts in the three 8 h time steps is independent from each other. In a first step, based on the probabilities P(0/0/1),  $P(0/\frac{1}{2})$  and P(1/3 / 1/3 / 1/3) (which sum up to  $\sum P=1$ ), the number of wet 8 h time steps is identified. The choice of the wet intervals among all three possible intervals.

15. Line 4, page 7. What does it mean that the (probability?) parameters depend on the volume? How? Even if this is explained in a previous paper, it should be briefly recalled here for the sake of clarity.

The following explanation has been added:

"Müller and Haberlandt (2015) have shown that for days with high rainfall amounts (above a quantile q0.998) the probability for two and especially three wet 8 h time steps is much higher than for lower daily rainfall amounts. Without a consideration of this volume-dependency of the parameters, the probability is too high that high daily rainfall amounts are put into one 8 h time step, which will lead to an overestimation of extreme rainfall values."

5

16. Lines 15-20, page 7 and 1-2, page 8. How many parameters characterize this modified version of the model? Which is the rule of dependence for starting, enclosed, etc. elements in the disaggregated series? The explanation is incomplete, or at least too vague.

- Regarding the parameter number, Table 3 has been added to the manuscript. The explanation has been extended by the 10 following sentences to describe the parameter estimation procedure more concise: "To summarize the previous explanation regarding parameter estimation: for each temporal scale two fine time steps are aggregated (or three finer time steps for b=3, respectively) to one coarser time step, whereby the position and the volume class of the coarser time step determines to which position-volume class-combination the current splitting belongs. The
- cascade model parameters are then estimated over all splittings of a position-volume class combination." 15

17. Section 3.2.1. Method B adds additional probability parameters, letting them vary with the "position". It is unclear how these multiplicative parameters are determined to reproduce the statistical characteristics of the process across scales based on the structure of the cascade (i.e. b=3 for level one and b=2 for all the remaining levels up to the desired temporal

20 resolution). See also previous comment.

25

I assume that this comment refers to section 3.1.2. Additional to the added explanation (see my reply to your previous comment) a summarizing figure (Fig. 3) has been added, including all position definitions applied in this study.

18. Figure 2 could be improved to help for reader understanding the difference between method B and C (but also starting, ending etc. elements of the cascade). Further, it should be explicitly mention if blue and white denote wet and dry states.

- We thank the reviewer for her suggestion. We added the temporal scales to Fig. 2. Since a new figure (Fig. 3) has been added to the manuscript, which illustrates in detail the different possible position classes, the colours have been removed from Fig. 2.
- 30 19. Line 11, page 9. "isolated" instead of the second "ending"? The reviewer is right, "ending" was replaced by "isolated".

20. There is a substantial difference between the approach proposed here and that used in Lombardo et al. (2012, 2017). Here the Author introduces a "conditional" probability that determines the probability of a wet or dry state, while in Lombardo et

- al. the information of previously disaggregated elements at the same time-resolution is used to feed a linear interpolation 35 model estimating the disaggregated value of the subsequent element (see eq. (7) in Lombardo et al., 2017). The reviewer is right, only the identification of the time steps "most worth to consider" has been overtaken from Lombardo et al. (2017), while the disaggregation method itself differs. The related part of the manuscript (P3L17-20) has been rephrased to:
- 40 "The second, more complex modification follows an idea of Lombardo et al. (2012, 2017). Lombardo et al. analysed which time steps are most worth to consider to generate highly correlated time series under the burden of computational efforts. Their conclusion is adapted in this investigation."

21. Line 7 page 10. The underestimation of the autocorrelation function characterizing the starting model is not a characteristic of all multiplicative random cascade models, as stated by the Authors in the literature review. Is there a motivation for this? Is it possible to generalize the problem to a specific MRC model (i.e. generator) structure? I'm not sure if the text index refers to the comment (this reply is to the comment itself).

- 5 First, the deviation of the autocorrelation in the generated time series from the observed time series refers to micro-canonical cascade models, not cascade models in general. This information has been added where possible in the introduction. Second, under- and overestimations are reported by the references in the introduction. But indeed, the previous results from the basic model for this study leads to underestimations of the autocorrelation.
- However, it is assumed that with the new position definitions and the resampling approach under- and overestimations of the autocorrelation function can be avoided and improved subsequently, respectively.

22. Line 17, page 14. If the reference model works well in terms of autocorrelation, that is the main issue here, why two different modifications are proposed?

In the referred sentence only the results for lag-1 autocorrelation are discussed, how it is done very often for rainfall characteristics. However, as shown later in e.g. Fig. 7 and related paragraphs, the analysis of lag-1 is not enough, the whole autocorrelation function has to be analysed, or at least more lags as done in this study for lag-6 and lag-36.

23. I do understand that the dataset is not so rich, but I expected to find two separated datasets, one for "calibration" and one for "validation" of the models. The Author should justify this choice.

- 20 The maximum time series length is 20 years. For the parameter-intensive method C an artificial shortening of the time series should be avoided. Also, for later "real-life" applications, the whole high-resolution time series is used for parameter estimation for the disaggregation of the daily time series. Hence, a shortening of the time series could lead to a worsening of the disaggregation results, which would not occur otherwise. The following explanation was added in the validation section 3.1.4:
- 25 "A split-sampling into calibration and validation period was not carried out to keep the time series as long as possible for the parameter estimation (see also the discussion in Section 3.1.4)."

**Review by anonymous referee, Referee #3**

**1** Summary**

5 The paper deals with the improvement of a given disaggregation model using microcanonical cascade. In general the topic is relevant for the community. The paper is interesting, but cannot be published in its current state, and requires major modifications.

Reviewer #3 is gratefully acknowledged for her/his efforts and the time spend on the manuscript. I think the comments of the reviewer refer to the original submitted version, not to the "current" version. The current version was uploaded later and

10 includes already the reviews of Elena Volpi (reviewer #2) and reviewer #1 and covers some of the issues and concerns raised by the reviewer. Nevertheless, please find below a detailed point-by-point reply under the assumption, that the review was based on the original submitted manuscript. All page and line numbers refer to the original submission.

**2. General comments:**

15

35

- The paper is quite hard to read with many models being compared. Explanations for the slight variations between the various models are sometimes hard to follow. There is a lack of mathematical details in the presentations of the various models.

The reviewer points out several issues about the method description and the complexity of the manuscript. These issues are repeated as detailed comments, hence please find there a detailed reply.

- Only comparison between variations of a given model are provided. Comparisons with other type of cascade models should at least be discussed.

I agree with the reviewers suggestion that the manuscript would benefit from comparisons of the disaggregation results with

- 25 results from other cascade models. However, most references compare only the value for the lag-1 autocorrelation, which is not representative (e.g. as shown for method C in Fig. 5 with an overestimation of the lag-1 autocorrelation, while in Fig. 6 a clear underestimation for the majority of lags can be identified). Hence, a comparison is hardly possible, especially because a comparison based only on lag-1 values could be misleading.
- Also, the autocorrelation depends on the rainfall processes and genesis in the study area, which enables comparison of absolute values from other studies. A fair comparison could only be carried out with the direct application of another cascade model, but this would be contrary to the reviewers comment on the already very large complexity of the manuscript with too much model version comparisons.

- There are numerous parameters to be estimated per model (not even very clear which number according to the model choice). It is not clear whether a calibration period and a validation period were used.

- To provide a better overview an additional figure and a table were added to the manuscript. Fig. 3 provides an explanation of position definitions for method A, B and C in dependence of the branching number. Table 2 lists the cascade model parameters for methods A, B and C in dependence of the applied branching number. From Table 2 it is visible, that for method C a high number of parameters is required. Since the maximum length of the observed time series is 20 years, no
- 40 split-sampling has been applied, to ensure a best-possible parameter estimation. Also, for later "real-life" applications, the whole high-resolution time series is used for parameter estimation for the disaggregation of the daily time series. Hence, a shortening of the time series could lead to a worsening of the disaggregation results, which would not occur otherwise. The following explanation was added in the validation section 3.1.4:

"A split-sampling into calibration and validation period was not carried out to keep the time series as long as possible for the

45 parameter estimation (see also the discussion in Section 3.1.4)."

**3. Detailed comments:**

1) Introduction

- p.2 l.21 : "since time series with 1280 minutes do not exist as observation do not exist". I do not understand this statement and this does not seem a real issue. Anyway, if needed, you can disaggregate at a higher resolution and up-scale to the desired one.

5 The aim of real-life applications is to disaggregate daily values to achieve high-resolution rainfall time series. The implication of putting a1440 min rainfall amount (1 d) into 1280 min and start the disaggregation is that you will end up with a day which is 160 min (~2.5 h) too short. So either you assume that the missing time steps have a rainfall amount of 0 mm (the question is then: Where to put them?) or you apply any transformation to the daily time series before the disaggregation. Both approaches will affect the resulting rainfall time series characteristics. The following sentence was added for clarification:

"Of course, by the application of an additional transformation process a desired temporal resolution can be achieved, whereby the transformation process affects the characteristics of the disaggregated time series."

**2) Rainfall data**

- p. 3 1.28-29 : "from a practical. . . have an impact on the autocorrelation function". Why not trying the compute the autocorrelation using higher moments to limit the influence of smaller values ?
 The reviewer suggests another way of computing the autocorrelation. Indeed, by doing so the influence of the too small intensities would be limited, but it solves not the problem itself. Also a threshold could be introduced to "ignore" the lower

- values. However, especially for method C very long wet spells with too small rainfall intensities are generated. This
   underestimation of the average intensity (Fig. 5) poses a serious problem for later applications. Hence, the introduction of MRA and MMD to avoid too small rainfall intensities is also done for practical applications, not only for a more representative computation of the autocorrelation.
- p.4 1.10 : "how can a minimum rainfall intensity be ensured during the disaggregation process?". It is not very clear to me
   the need for this, since as pointed out by the author and references cited, it might very well be simply due to the rain gauge measurement limitations. It might be worth testing a time series obtained with a disdrometer which enables better representation of small values of rainfall.

Indeed, a disdrometer time series would gain more insights in this relation of observed and disaggregated rainfall intensities, but is not available to me for this study area. However, the long wet spells with very low rainfall intensities generated by method C have to be avoided (please see also my reply to your previous comment). This is the motivation for the

30 method C have to be avoided (please see also my reply to your previous comment). This is the motivation for introduction of MRA and MMD (see P14L25-32).

3) Methods - p.5 1.8 : "actual" - - > "Actually"

35 This has been corrected.

- Eq. 1 : it should be added how lambda is related to t1 and t2 (I guess lambda = t2-t1) This has been added in the paragraph before Eq. 1.

40 - p. 6 l.24 : "on" - - > "no" ? This has been corrected.

- Eq. 2 : shouldn't it be P(0/0/1)/3 in the first line since there are three possibilities (100,010,001) for the same P(0/0/1)? This remark is also valid for all the other probabilities except P(1/3,1/3,1/3)

45 The reviewer is right, the way Eq. 2 was written was misleading. Eq. 2 has been changed to:

$$W_{1}, W_{2}, W_{3} = \begin{cases} \{1, 0, 0; 0, 1, 0 \text{ or } 0, 0, 1\} & \text{with } P(0/0/1) \\ \left\{\frac{1}{2}, \frac{1}{2}, 0; \frac{1}{2}, 0, \frac{1}{2} \text{ or } 0, \frac{1}{2}, \frac{1}{2}\right\} & \text{with } P(0/\frac{1}{2}/\frac{1}{2}) \\ \frac{1}{3}, \frac{1}{3}, \frac{1}{3} & \text{with } P(\frac{1}{3}/\frac{1}{3}/\frac{1}{3}) \end{cases}$$

- p.7 1.11 : "an empirical function", please be more specific (see also general comment on the lack of mathematical details). The description of the empirical distribution function was extended to:

5 "An empirical distribution function is used to represent f(x), with a maximum of 14 equidistant bins (based on the number of available splittings, see Storm (1988, p. 86))."

- p.7 l.15-18 : a summary table or scheme would be helpful.

Fig. 3 has been added to illustrate all position definitions.

10

- It remains weird to have different branching number and probabilities weights for the first cascade steps which seems to be in contradiction with the underlying scaling properties.

The different branching numbers are a compromise between scaling theory and practical applications. Indeed, it can be questioned if scaling properties are met by b=3 and especially by the uniform distribution in this first disaggregation step.

Although the number of wet 8 h-time steps is realistic, the resulting rainfall amounts are maybe less realistic. There are two 15 reasons for this assumption.

First, the number of wet 8 h-time steps tells us something about the genesis of the rainfall event. If it is only one wet time step, it is likely a convective event and hence the whole daily rainfall amount is put into this single 8 h-time step. If the rainfall event lasts longer than 8 h, it is likely a stratiform event, so a long-lasting and less intense event (with often more or

less uniform rainfall occurrence). The uniform distribution can be found only on the 8 h-level, on finer temporal resolutions 20 intensities vary due to the b=2-splitting so the resulting final time series does not show any uniform rainfall intensity distribution anymore.

Second, the assumption with the uniform distribution demands only a few parameters. For method A, two parameters are required (P(0/0/1)) and P(0/0.5/0.5)). The only other tested approach with a b=3-splitting in the first disaggregation step was

- introduced by Lisniak et al. (2013), who use for example 8 distribution functions for a splitting with 3 wet 8 h intervals. To 25 keep the cascade model parameter parsimonious, the uniform distribution has been chosen. Also, in previous publications (Müller and Haberlandt, 2015, 2018, Müller-Thomy et al., 2018, Müller-Thomy and Sikorska-Senoner, 2019) with this assumption good representation of rainfall characteristics have been achieved.
- 30 - Section 3.1.3 : I found the paragraph quite hard to read. may be a more precise scheme could be helpful. It should be mentioned that it adds a lot of parameters. In general, a summary table with the number of parameters according to the model would be helpful.

The existing scheme in Fig. 2 was improved and Table 3 including all parameters in dependence of the applied branching number was added as well as section 3.1.4.

35

- Section 3.2 : why presenting two different models (especially given that they provide rather similar results) ? It adds complexity to a paper with already a lot of comparison. I would keep only the MMD which is the more realist I believe. This suggestion was also made by reviewer #1. He suggested as well to leave out the MRA approach to reduce the

40

complexity of the manuscript. However, both approaches represent possible solutions and none of them has been tested before to the authors knowledge. To reduce the complexity, only the MMD approach has been applied for the resampling investigations. I prefer to keep both approaches in the current study, especially since reviewer 2 identifies this issue as the more fundamental one. The comparison of the two approaches is also useful for other researchers, because they know that the outcome is very similar and don't have to carry out a study on their own.

- Section 3.3 : The process with Ir and more generally the swapping seems rather ad hoc. It seems that the underlying physical meaning of cascade process is lost. I think that this issue should at least be discussed.

Maybe the swapping seems ad hoc because it is not related to cascade models in general. However, in a previous publication Müller and Haberlandt (2018, Fig. 12) have shown, that the scaling behaviour of the disaggregated time series is not changed by the resampling process. The following sentence has been added:

"As proven by Müller and Haberlandt (2018), the resampling does not affect the scaling behaviour of the disaggregated time series, because the total rainfall amount as well as the number of wet time steps are kept."

**- p. 13 l. 18 : "30 realisations". Why such a small number, it seems that much more could have been performed.**

10 The disaggregation is a random process. In a prior study it was analysed, which number of realisations is required to cover the stochastic behaviour of the disaggregation based on the rainfall characteristics of the disaggregated time series analysed in this study. It was found that 30 realisations are sufficient. Also, as pointed out as last point in section "5.3 Study limitations", the simulated annealing is an optimization process that demands a high computational effort. Hence, 30 realisations are kept for the study.

15

5

4) Results

- Table 3 and 4 are really hard to follow. I think a scheme representing the various cases could be really helpful. A scheme of all possible positions is provided in Fig. 3 for a better interpretation of both tables.

- 20 Why the average rainfall intensity changes is such a micro-canonical cascade ? I'm not sure if I understand the question correctly. The average rainfall intensity depends only on the number of wet time steps generated by the cascade model, since the total rainfall amount remains the same for the whole daily time series. Hence an overestimation of wet time steps leads to an underestimation of the average intensity and vice versa.
- p. 19 l. 5-10 : may be a graph showing the sensitivity of the results to Ir would be be needed.
   The reviewer is right, this would be useful. However, a 2D-graph could only include the results for one lag and one extreme event (of a certain duration and return period). Since the problem is more complex, a simplification as a 2D-graph was not added to the manuscript.
- 30 5) Discussion p. 21 l. 14-15 : "identified similarities. . . used for simplification", please clarify and be more explicit. The reviewer is right, due to the high number of parameters every possibility to reduce their number should be listed in detail. The following sentence has been added: "
[revised manuscript text omitted]

---

## Editor Decision (ED1)

[revised manuscript text omitted]

---

## Author Response (AR2)

**Temporal rainfall disaggregation using a micro-canonical cascade model: Possibilities to improve the autocorrelation**

Please find in this document the detailed reply to the reviewer comments as well as the manuscript with tracked changes afterwards.

**Reviewer #2**

The Author has addressed all the comments provided by the Reviewers in the point-by-point reply to Reviewers comments; he clarified some aspects related the scope of this work and the methodology implemented to improve the general discrete multiplicative random cascade structure. However, some major and minor questions are still open and are not fully addressed in the revised manuscript.

I thank the reviewer for pointing out the improvement resulting from the previous review round. A short summary of my changes in the current review round: Additional investigations regarding scaling function and leave-one-out cross-validation were carried out and added in the manuscript. However, since it was mentioned in the first review round that the content of the manuscript is too excessive (hence MRA-MMD comparison was left out), additional analysis regarding the non-stationarity issue of multiplicative cascade models were not carried out. As suggested by the reviewer, it is clearly stated that this problem exists and that it is not tried to solve it in this study. Please find a detailed point-by-point reply to your comments below.

1. The major gap of this work is, based on my opinion, the lack of consideration of the non-stationarity property of MRC models. Which is the theoretical and practical relevance of a model that reproduces a "uncontrolled" non-stationary process?
The reviewer questions the usage of the disaggregation model with its current limitations. Indeed, the non-stationarity is not solved in this study, but that was also not the aim. The cascade model in the current study with this limitation is applied successfully in the fields of rainfall-runoff modeling (Müller-Thomy et al., 2018, Müller-Thomy & Sikorska-Senoner, 2019), urban hydrology (Müller & Haberlandt, 2018) or erosion processes (Bezak et al., in preparation), whereby success refers to the generation of rainfall time series with a good representation of the rainfall characteristics needed in these study fields. Based on the successful applications it seems that the "uncontrolled" non-stationary process plays only a minor role (at least for the investigation areas and applications of the aforementioned studies). That means not that I do not see the need to solve that issue, but from a practical point of view the application of the cascade model in its current form is still feasible.

Changes to the model structure proposed in this work allow to better reproduce the autocorrelation function only on average (in time), since the autocorrelation function is not the same for each time step, i.e. it depends on the position in time at the fine resolution of cascade simulation; so, what happens at each time step? Ensemble simulation and autocorrelation estimation could help to clarify this issue. Hence, based on my opinion the Author should first fix the stationarity issue and then think about improving the stationary autocorrelation (independent from time to time) reproduction. It is also possible that the proposed model improvement already has an effect on the theoretical stationarity of the simulated process. If the Author does not want to tackle this complicated issue, he should at least mention and discuss it explicitly in the introduction of the revised manuscript, so that potential readers are aware of the important limitation of the proposed model.

The reviewer suggests to either implement additional investigations regarding the non-stationarity issue or to discuss it in the introduction to raise the awareness of the potential reader. Since the non-stationarity issue is not the topic of the manuscript, the following sentences were implemented in the introduction:

"It should be pointed out that Lombardo et al. (2012) show that the disaggregation process of discrete multiplicative random cascade models is non-stationary, "because the autocorrelation structure depends on the position in time in an undesirable manner". Since the aim of this study is to improve the overall representation of the autocorrelation function (as average over time), it is not analysed to which extent the non-stationarity issue is solved by the introduced methods."

2. A second major gap is the lacking of the derivation of the theoretical autocorrelation function given the modifications applied to the cascade, such as e.g. the different branching number at the first and subsequent levels etc. The dependence of the theoretical autocorrelation function on the model parameters (including the branching number) could help the reader understand the complexity of the model and the parameters that need to be calibrated.

5 The reviewer suggests to implement a comparison between the theoretical autocorrelation function of the introduced cascade models. In my opinion the analysis of the theoretical autocorrelation would be part of the (non-)stationarity analysis mentioned in the major comment before and represents not a second major gap. I fully agree that this analysis would be required if the model will be developed to overcome the non-stationarity issue. However, since it is clearly stated in the current version of the manuscript that this is not the study intention (see reply above), I did not implement such a
10 comparison. In this study the autocorrelation is compared with the empirical autocorrelation resulting from observations, which provides enough insight to the results and for the study aims. The derivation of the theoretical autocorrelation function would again require its own subsection with mathematical explanations (or extensions of the cascade model descriptions).

Further, it is not clear if the scaling (fractal or multifractal) property that is at the core of MRC model is conserved by the
15 modifications introduced to the model. A theoretical comparison with the classical MRC model, that could be recalled at the beginning of the method section, would help for this.

I thank the reviewer for this suggestion. However, if the reviewer refers by "classical MRC model" to a micro-canonical cascade model with b=2 in all disaggregation steps, fine-graining to a very high temporal resolution would be required to coarse-grain it back to 5 min temporal resolution. The parameters for that cannot be estimated from the observed time series
20 in this study, so a comparison is not possible. Also, it would bring another cascade model variant into the manuscript, which would be only used for this comparison. Since a comparison as suggested by the reviewer is not possible in this study, please find a comparison of the scaling behaviour of observed and disaggregated time series below.

The scaling behavior of the observed and disaggregated time series was analyzed by

$$M_q = \lambda^{K(q)}$$

25 with moments M, moments order q, the moments scaling exponent K(q) and the scale ratio λ. The results are similar among the stations and for the moments q=1, q=2 and q=3 (only these have been analysed). As shown in Fig. 1 the scaling behaviour of the observations is represented well by the three disaggregation methods A-C. While method A and B show very similar results, method C shows slightly different results. The deviations for the high-temporal resolutions maybe artefacts of the linear transformation process to achieve a final temporal resolution of 5 min. Nevertheless, since both axes
30 are logarithmic, the deviations are negligible small.

[Figure]

Fig. 1: First moment q=1 for station Cuxhaven, single values for each method represent the mean of 30 realisations (results are for other stations and moments similar)

35 The method for estimating the scaling behaviour was implemented in the evaluation section of the disaggregated time series (Sect. 3.4), the results are interpreted at the beginning of Sect. 4.1.

3. The definition of the generator of the MRC is in the classical MRC models generally requires to introduce a probability model for the random variable W (the multiplicative weight). Can the Author explicitly define this probability model? If it is described by the probability values P in Eqs. 2 and 3 or 4 and 3 (it is an empirical distribution function), this means that each of the probability value (minus one thanks to the sum equal to 1) is a free parameter that needs to be calibrated from the observation.

The reviewer is right, Eq. 2 and 3 describe the probability model and only empirical distribution functions are applied. To clarify this vague information, we have added the information at the end of the general model description in Section 3.1.1 (three paragraphs before 3.1.2):

"The cascade model parameters are then estimated over all splittings of a position-volume class combination. " -> "The cascade model parameters are then estimated over all splittings of a position-volume class combination, so all parameters and distribution functions included in the disaggregation process are estimated empirically."

Further, parameters seems to be doubled since they are estimated based on two volume classes, separated by the 0.998 quantile of daily total amounts. Given this high threshold, the number of values used to calibrate the parameters of the lower class are expected to be much larger than those pertaining to the upper class, so that the uncertainty characterizing the two classes of parameter values are strongly different. Further, the threshold of separation of volume classes changes from the first to the subsequent disaggregation levels and then with the position. Are those considerations correct?

The considerations of the reviewer are correct.

Are the parameters summarized in table 2 all free parameters that need to be calibrated? In such a case, the model is strongly over-parametrized with respect to classical MRC models that are characterized by a few parameters. Indeed, the fact that the parameters are estimated from observations means that they are calibrated to reproduce the observed realization of the process. What happens if the model calibrated on the available dataset is applied and tested against a different dataset (another possible realization of the same stochastic process)? Since the model is over-parametrized it is strongly needed to test it for out-of-sample observation. This is also a very important point, that was not clear in the first version of the manuscript and that needs to be addressed before the manuscript can be considered for publication in HESS.

An additional out-of-sample analysis was carried out via leave-one-out cross validation. Aggregated daily values of one station (target) were disaggregated with parameters estimated with the time series of the closest station (neighbour). The average distance between 'target' and 'neighbour' station was 38.9 km. The analysis and the results are attached in Appendix 1 one of the manuscript (and referenced in the manuscript). The resulting rainfall characteristics (Table A1) and autocorrelations (Fig. A1) of the disaggregated time series are very similar for both parameter sets, estimated at the target or at the neighbouring station. This confirms the model parameter transferability for the study area.

4. As for the branching numbers chosen in this model, I would avoid to introduce a "physical" explanation of the assumption of 8-hours wet/dry periods, since it is quite arbitrary the choice of this temporal duration. However, if the Author has a clear reference that justify this choice independently of the climatic location, this should be cited.

The reviewer argues a missing physical reasoning for the b=3-splitting and asks for references as justification. As mentioned in the introduction and method description, the b=3 splitting is only introduced to achieve a final temporal resolution of 5 min, there is no physical reasoning for it (and this is also not stated in the manuscript). Nevertheless, an interpretation of the parameters is possible as it would also be the case for b={4,5,6,…,n}. The parameters for b=3 describe only i) rainfall occurrence and non-occurrence, and ii) at which time step the rainfall occurs (over these time steps the rainfall amount is distributed uniformly). From a logical point of view it is unlikely that the rainfall at days with the highest rainfall amounts (based on q>0.998 as threshold) falls in the first (hour 1-8) and the third (hour 17-24) interval with no rainfall in between. Thus, the physical interpretation of the parameter values was not removed. Despite the fact that rainfall generating processes can differ during the day and can for some cases depend on the time (e.g. land-sea winds), I have implemented the following sentence in the general model description (Section 3.1.1, second paragraph) to clarify that this was not the intention in the current study:

"The choice of *b=3* has no physical reason and is only applied to achieve a final temporal resolution of 5 min (see Section 1)."

Some specific comments follow, where page and line numbers refer to the revised version of the manuscript.

*Minor comments:*
Abstract. Since it is recognized that MRC can underestimate or overestimate the autocorrelation function, the Author should
be more general in the abstract mentioning both the problems.
Thanks for pointing it out, it is corrected:
„underestimation of the autocorrelation" -> „inadequate representation of autocorrelation"

Further, method C is mentioned in the abstract before its definition; please fix it.
Thanks for pointing it out, the nomenclature was implemented in the abstract for method A, B and C.

Please specify if resampling is only an alternative method for improving autocorrelation estimation with respect to basic
MRC models or if it is introduced here for a different scope.
The following sentence was added to the abstract for clarification:
„Simulated Annealing as a post-processing strategy was tested as both, as an alternative as well as an addition to the
modifications in method B and C to improve the autocorrelation."
In the manuscript itself, it is stated p4L11-12: „In addition to the modifications of the cascade model itself, a resampling
algorithm as post-processing strategy is analysed to improve the autocorrelation"

P 2. L. 23-25. Since this is the common practice, the Author could be more specific on the transformation that alter the
theoretical properties of the target resolution rainfall time-series. As it stands, the comment is still vague.
I thank reviewer #1 for pointing out the vague information: The information is mentioned now more explicit:
„However, the majority of investigations with cascade models focus on the disaggregation of quasi-daily time series (with
time step durations of 1280 minutes instead of 1440 minutes, ...) down to 10 minute or 5 minute time series, which has the
advantage of using the same branching number of $b=2$ throughout the disaggregation process, that determines the number of
finer time steps (here: two) with equal duration from one coarser time step."
was reformulated to:
„However, the majority of investigations with cascade models focus on the disaggregation of quasi-daily time series (with
time step durations of 1280 minutes instead of 1440 minutes, ...) to achieve a final temporal resolution of 10 minutes or 5
minutes. This enables using the same branching number (that determines the number of finer time steps with equal duration
resulting from one coarser time step) of $b=2$ throughout the disaggregation process with intermediate resolutions of {640,
320, 160, 80, 40, 20, 10, 5 min}."

p. 3, l. 8-10. The point here is to what extent theoretical properties are reproduced numerically; it also depends on how
numerical assessment is performed, if by ensemble or time-series simulations. Further, it is expected that higher the number
of disaggregation levels the better is the accuracy of the model in reproducing the autocorrelation function (if the multifractal
character holds true across all the scales).
The reviewer refers to the sentences:
„A good representation of the lag-1 autocorrelation was achieved by Hingray and Ben Haha (2005) with two micro-
canonical cascade models. However, since only four disaggregation steps were applied (from hourly to 7.5 min time steps) it
remains unclear, if the good representation would have been achieved for more disaggregation steps."

I'm not sure in which direction the reviewer points with the numerical details in the reproduction and the assessment of the
disaggregation products. Also, it remains unclear for me why the reproduction of the autocorrelation should become better
with more disaggregation steps. The daily values result from observations and hence represent the original autocorrelation.
Hence, the autocorrelation cannot be improved with additional disaggregation steps to finer temporal resolutions, it can only
be as good as at the daily scale.

However, since the comments of reviewer #1 are very constructive and beneficial for the manuscript, I'm afraid of not being able to implement this comment. Maybe reviewer #1 can be a bit more explicit in the next review round, I did not change anything so far in the current version of the manuscript regarding this comment.

5    p. 8, l. 9-11. Bounded cascades allow the multiplicative weights W to depend on the cascade level and converge to unity as the cascade proceeds, so that the simulated random process becomes smoother on smaller scales (that's the meaning of "bounded"). In the literature, bounded random cascades have been applied to the stochastic fine graining of rainfall observations into high resolution data both in the canonical and micro-canonical form (e.g. Menabde et al. 1997). Hence, bounded cascade are not introduced to reproduce the multifractal behavior, since this is reproduced even using unbounded

10   cascades. Please clarify which is the rationale and the main objective of bounded cascades according to previous literature.
The reviewer is right, the explanation was misleading. The intention was to mention that on different scales different processes influence the rainfall occurrence (so multi-process instead of multifractal). The reasoning for the application of bounded cascade models was corrected and adopted to the explanation of the reviewer, the recommended reference was implemented:

15   "In bounded cascade models the weights $W$ depend on the temporal resolution to allow the disaggregation process to become smoother with finer resolutions (e.g. Menabde et al., 1997). The need for particular parameter sets for each temporal resolution arises from the wide temporal range (from daily to 5 min time steps) and hence the underlying processes, which differ between the temporal scales."

20   *Note: I assume by the notation that the reviewer changed the documents for the following comments (the minor comments before referred to the manuscript itself, while the notation from here on refers to the pdf including the authors reply in before. I just want to mention it for the sake of completeness and hope that my replies refer to the correct parts.*

   p. 21, l. 1-2. Only the validation or also the calibration of the model used all these information/summary statistics?

25   These rainfall characteristics are only used for the validation. The calibration is based purely on the aggregation of 5min precipitation time series. However, all 24 stations have been used in the study.

   p. 22, l. 7. Which optimization?
The Simulated Annealing is an optimization algorithm. The sentence before has been extended for clarification:

30   "...later introduced resampling algorithm..." -> "...later introduced resampling as optimization algorithm..."

   p. 30, l. 3. The term "diurnal cycle" is introduced here for the first time. What does the Author mean with this term and what does he refer to? This could be mentioned before, at the beginning of the methodology section.
The reviewer points out the missing explanation of the term "relative diurnal cycles". The following sentence was added:

35   "In a relative diurnal cycle the diurnal cycle with absolute rainfall amounts per time step is transformed by dividing the single rainfall amounts by the total rainfall amount of that day."
This explanation has been added behind this sentence, because the relative diurnal cycles are only applied in the resampling algorithm. A mentioning before this subsection would be misleading.

40   p.30, l. 18. I suggest to use "quantities" or "statistical summary" instead of "parameters" to avoid confusion.
I thank the reviewer for this suggestion and have applied it as recommended.

   p.30, l. 23-24. Which is the scaling behavior mentioned here? This is not introduced and mathematically formulated in the manuscript. This is related also to the mathematical formulation of the autocorrelation function as requested in my previous

45   major comment 2.

The reviewer refers to the sentence: "As proven by Müller and Haberlandt (2018), the resampling does not affect the scaling behaviour of the disaggregated time series, because the total rainfall amount as well as the number of wet time steps are kept."

The mentioned scaling behaviour is now part of this study, so its mentioning in the resampling section should be clear now (Sect. 3.4, which includes the description, is also referenced).

p. 33, l. 21-24. Since n=30 is a unexpected small number, I'm wondering if the convergence of the stochastic properties is tested on 30 realizations after averaging in time or not (i.e. for each time step or specific time quantities). Please clarify this issue.

Reviewer 1 is worried about the number of realizations. Indeed, this number seems to be chosen individually between different authors (see examples in Table 1 of the reviewer response.). Of course, additional examples for both, more or less realizations, can be found. In my study, the stochastic properties were determined for each disaggregated time series. Plotting the mean and the resulting range of the stochastic properties against the number of realisations leads to an asymptotic behaviour for 30 realisations and more.

The text was rephrased to:

"By 30 realisations the random behaviour of the disaggregation process is fairly well covered. The mean and the range of the event-based and continuous rainfall characteristics were plotted against the number of realisations used for their estimation, and an asymptotic behaviour was identified with increasing numbers of realisations after 30 realisations."

Table 1: Number or realizations used in disaggregation processes for different temporal resolutions

| Number of realizations | Target resolution | Author(s) |
|---|---|---|
| 10 | 1 hour | Olsson (1998), Pui et al. (2012) |
| 12 | 1 hour | Rupp et al. (2009) |
| 100 | 1 hour | Güntner et al. (2001) |
| 100 | 1 hour | Lisniak et al. (2013) |
| 10 | 15 minutes | Jebari et al. (2012) |
| 50 | 10 minutes | Paschalis et al. (2014) |

Also, the results indicate clear results with significant differences between the disaggregation approaches. It is assumed, that an increasing of the number of realizations will not change the results.

**Reviewer #3**

I already reviewed the initial version of the manuscript. The readability of the paper has been significantly improved. With regards to my comments (and the other reviewers' ones as well), it seems that the author choose not to implement most of the suggested additional work, notably with regards to actual comparison with other cascade models, and only to focus on the improvement of the discussion on the comparison with the other models. I my mind, this unfortunately limits the general scope / interest of this present study, without preventing it from being published.

I thank reviewer#3 for the recognition of the improvements. Indeed, a comparison with other rainfall disaggregation models would be interesting. However, the comments of reviewer #1 demand a deeper analysis of the improvement of the current disaggregtion model in this study. An additional comparison with other models would be too much content for one manuscript, also the focus would get lost. I decided for the in-depth analysis, but keep in mind the model comparison for further analysis.

[revised manuscript text omitted]

---

## Author Response (AR3)

**Editor**

I thank the editor for his additional remarks. The majority of the remarks has been addressed and are not replied here point-by-point, because most of them were rephrasing and shortening issues. All changes can be found in the tracked-changed version below. The few remarks which were not addressed are discussed below.

P6Eq1 Shouldn't it be: r_{t1,t2}(Lambda)? Is the correlation depends on the lag?
Lambda=t2-t1, so it is implicitly involved. Although including lambda instead of t1 and t2 seems more intuitive, this is the standard formulation in text books. I prefer to not change it.

P7l15 Where is V in eq. (3)?
The rainfall volume of the finer time steps is obtained by multiplying the rainfall volume $V$ of the coarser time step with the weights $W$. Eq. 2, 3 and 4 describe only how the $W$'s are determined. There is no additional equation which includes $V$, since the multiplication itself is trivial.

P7l22 99.8th percentile (instead of quantile0.998)
From my understanding percentiles can only be integer values, so I prefer to keep the quantile term, but extended it to empirical quantile to specify it.

P17 Fig6 Should be in log-log scale?
Yes, the figure should be in log-log scale as pointed out on P15L10.

P21l11 "approx."
I'm confused by the highlighted term "approx.". For the case that this abbreviation is not well-known, I have extended it to approximately.

P28l5 Should indicate here which of the method is better.
There is no universal recommendation and it depends on the intended application and the therefore important rainfall characteristics. Hence, I recommend the leave the conclusion as it is.

[revised manuscript text omitted]